# Evolutionary and functional insights into the mechanism underlying body-size-related adaptation of mammalian hemoglobin

**Olga Rapp, Ofer Yifrach\***

Department of Life Sciences, Zlotowski Center for Neuroscience, Ben-Gurion University of the Negev, Beer Sheva, Israel

**Abstract** Hemoglobin (Hb) represents a model protein to study molecular adaptation in vertebrates. Although both affinity and cooperativity of oxygen binding to Hb affect tissue oxygen delivery, only the former was thought to determine molecular adaptations of Hb. Here, we suggest that Hb affinity and cooperativity reflect evolutionary and physiological adaptions that optimized tissue oxygen delivery. To test this hypothesis, we derived the relationship between the Hill coefficient and the relative affinity and conformational changes parameters of the Monod-Wymann-Changeux allosteric model and graphed the 'biophysical Hill landscape' describing this relation. We found that mammalian Hb cooperativity values all reside on a ridge of maximum cooperativity along this landscape that allows for both gross- and fine-tuning of tissue oxygen unloading to meet the distinct metabolic requirements of mammalian tissues for oxygen. Our findings reveal the mechanism underlying body size-related adaptation of mammalian Hb. The generality and implications of our findings are discussed.

DOI: https://doi.org/10.7554/eLife.47640.001

**\*For correspondence:**
ofery@bgu.ac.il

**Competing interests:** The authors declare that no competing interests exist.

## Introduction

Two major obstacles hamper understanding the evolutionary origins of properties seen in modern-day proteins. One stems from the fact that mutation and selection are separated by several levels of biological organization such that it is difficult to elucidate the complex relationships between protein sequence and organismal/population fitness. Second, even in those cases where the molecular properties of a protein (e.g. affinity or cooperativity) have adaptive value with respect to organismal fitness, we often lack an understanding of the molecular mechanism underlying changes in the values of these properties. Given these challenges, we currently understand the mechanisms underlying adaptive behavior for only a very limited number of proteins (e.g. *Mello and Tu, 2005*; *Keymer et al., 2006*; *Swem et al., 2008*; *Storz et al., 2009*).

Hb represents an excellent model protein to study molecular adaptation in vertebrates (*Perutz, 1983*; *Poyart et al., 1992*; *Hourdez and Weber, 2005*; *Storz and Moriyama, 2008*). Hb is a tetrameric allosteric protein responsible for tissue oxygen delivery in a manner controlled by the midpoint and slope of its oxygen saturation curve. These macroscopic parameters reflect the effective affinity ($p_{50}$) and cooperativity ($n_H$) of oxygen binding by Hb, respectively and are traditionally evaluated using the Hill equation, derived assuming an all-or-none mode of binding (*Hill, 1910*). Over the years, many studies have indicated that Hb oxygen affinity ($p_{50}$) is an adaptive trait, that is its value has been tuned during evolution so as to tailor tissue oxygen delivery to the particular metabolic requirements of vertebrates due to variations in body size, lifestyle and/or environmental conditions. For example, vertebrates native to mountain habitats where low ambient oxygen pressures

**eLife digest** In humans and other mammals, a protein in the blood called hemoglobin carries oxygen from the lungs to other parts of the body. This protein contains four subunits that can each bind to one molecule of oxygen, so a single hemoglobin can carry up to four oxygen molecules at the same time. Previous studies have found that, although each subunit can potentially bind oxygen on its own, the subunits actually work together to help each other bind to oxygen in the body.

Two biochemical properties of hemoglobin affect how it carries oxygen molecules. First, oxygen-binding affinity, or how tightly the protein can bind to oxygen; and secondly cooperativity, or the degree to which the subunits interact with each other to bind oxygen more tightly. Mammals of different shapes and sizes have different requirements for transporting oxygen from the lungs to organ tissues, which have shaped their hemoglobin proteins over evolutionary timescales. While the contribution of affinity to hemoglobin evolution in animals of different sizes has been addressed in the past, the role of cooperativity in hemoglobin adapting to body size has remained unclear.

Here, Rapp and Yifrach used a mathematical approach to analyze existing data from 14 different mammals – including mice, sheep, humans and elephants – on how oxygen binds to hemoglobin. Using this approach, they were able to explain why different mammalian hemoglobins present different oxygen-binding affinity and cooperativity values. Furthermore, they demonstrated that the cooperativity values were very close to the maximum they could be for each version of hemoglobin.

These findings suggest that, as mammals evolved, genetic mutations that altered the oxygen-binding affinity or the ability of hemoglobin subunits to cooperate may have allowed hemoglobin proteins to adapt to meet the oxygen needs of mammals of different sizes and shapes. In the future, the approach used by Rapp and Yifrach could be adapted to study how other proteins that bind molecules in a cooperative manner have evolved.

DOI: https://doi.org/10.7554/eLife.47640.002

prevail usually possess Hb that exhibits higher affinity for oxygen than does the Hb of lowland dwellers (*Lenfant, 1973*; *Bunn, 1980*; *Weber, 2007*; *Storz and Moriyama, 2008*; *Storz, 2016*). Likewise, Hb in small-sized animals that display high metabolic rates presents lower affinity for oxygen than does the same protein in large-sized animals, again reflecting the adaptive adjustment of tissue oxygen unloading by Hb (*Schmidt-Neilsen and Larimer, 1958*; *Schmidt-Nielsen, 1970*). Accordingly, early respiratory system physiologists assigned great significance to the $p_{50}$ parameter, terming it 'oxygen unloading tension', and proposed that Hb function could be explained in terms of adaptive evolution of this value (*Schmidt-Neilsen and Larimer, 1958*; *Schmidt-Nielsen, 1970*). At the same time, such studies ignored the adaptive potential of cooperativity ($n_H$) with respect to Hb function. This is surprising, given that $n_H$ also affects Hb tissue oxygen delivery. Moreover, mechanisms underlying changes in $p_{50}$, such as changes in oxygen affinity of the oxy- or deoxy-states of Hb, changes in the ratio between these states or changes in their sensitivity to allosteric effectors, are all also expected to affect $n_H$. Indeed, the finding by *Milo et al. (2007)* that mammalian Hbs exhibit variations in the Hill cooperativity value hints at the adaptive potential of the $n_H$ trait.

The need to also consider $n_H$ in addressing adaptation of Hb function becomes evident when we examine the implications of accepting the notion that molecular adaptation of Hb function in vertebrates relies primarily on its $p_{50}$ affinity property. Given the inherent tradeoff between arterial $O_2$ loading and peripheral $O_2$ unloading, it is not clear whether improved tissue oxygen delivery is achieved by increased oxygen affinity to improve Hb loading or by decreased affinity to favor unloading (*Storz, 2016*). Either very high or very low oxygen affinity would result in a low fraction of tissue oxygen delivery unless cooperativity ($n_H$) between Hb subunits is introduced as a means to switch between low- and high-affinity Hb states. It would thus seem that the values of both $p_{50}$ and $n_H$ of Hb should reflect physiological optimization of the tradeoff between $O_2$ loading in the lungs and $O_2$ unloading in tissues, so as to optimize tissue oxygen delivery. Although this is a seemingly obvious claim, it has yet to be tested experimentally.

Here, we tested whether the values of both $p_{50}$ and $n_H$ of Hb are indeed optimized in the case of body size-related adaptation of mammalian Hb, relying on extensive evolutionary and physiological oxygen saturation datasets available for mammalian Hb proteins. The former comprises a set of

oxygen binding curves of homologous Hb proteins from different mammalian species, all measured under similar physiological conditions (*Milo et al., 2007*; *Rapp and Yifrach, 2017*). The latter comprises a set of binding curves of human Hb measured under a variety of different physiological conditions, that is, in the presence of different concentrations of H⁺, $CO_2$ and organophosphate, all inhibitory allosteric ligands of Hb (*Rapp and Yifrach, 2017*). For all data sets curves, reliable estimates for $p_{50}$ and $n_H$ Hill parameters and for the elementary relative affinity (*c*) and conformational changes (*L*) parameters of the MWC concerted allosteric model (*Monod et al., 1965*; *Rubin and Changeux, 1966*) are available. By deriving the explicit mathematical dependence of both $p_{50}$ and $n_H$ on the MWC parameters of a tetrameric protein, we were able to draw the theoretical 'biophysical Hill cooperativity landscape' at half-saturation that graphically describes how $n_H$ varies with changes in the values of the MWC parameters. The mapping of experimentally derived Hill and MWC parameters of all evolutionary and physiological dataset curves on this landscape provided insight into the mechanism underlying the adaptive behavior of $p_{50}$ and $n_H$ with respect to tissue oxygen delivery by mammalian Hb and their relation to organism fitness. The principles underlying the 'biophysical Hill landscape'-mapping strategy are general and offer mechanistic-level understanding of how physiological and evolutionary variations operate to shape the molecular property of a protein.

## Results

### Hemoglobin evolutionary and physiology datasets

The evolutionary and physiological datasets for Hb used in the current study were respectively compiled in the meta-analysis of Hb function by *Milo et al. (2007)* and by *Rapp and Yifrach (2017)*. Specifically, the Hb evolutionary dataset comprises 14 different mammalian Hb oxygen saturation curves obtained under similar physiological conditions (Materials and methods and *Supplementary file 1*). The Hb physiological dataset comprises 26 different human Hb saturation curves collected under a variety of different physiological conditions, including those involving changes in H⁺ (*Di Cera et al., 1988*; *Imai, 1983*), $CO_2$ (*Doyle et al., 1987*) and 2,3-BPG (bis-phosphoglycerate) (*Benesch et al., 1971*) concentrations.

For each dataset curve, the magnitude of cooperativity in oxygen binding was evaluated using either the Hill (*Figure 1a*) or MWC model (*Figure 1b*). In the former, the sigmoidal saturation curve is fitted to the Hill equation (*Equation 1*) (*Hill, 1910*):

$$\bar{Y} = \frac{[S]^{n_H}}{[S]^{n_H} + K_{0.5}^{n_H}}$$

(1)

where $\bar{Y}$ represents the fraction of bound sites and $p_{50}$ (~$K_{0.5}$) and $n_H$ correspond to the midpoint and slope parameters of the curve. The Hill all-or-none binding mode corresponds to a simplification that is far from what occurs in reality. Instead, Hb oxygen binding generally involves intermediate ligation and conformational species, as delineated by the MWC allosteric model (*Marzen et al., 2013*; *Monod et al., 1965*; *Rubin and Changeux, 1966*) (*Figure 1b*) (see Appendix 1 for a detailed discussion on why we think the classical MWC allosteric model adequately describes hemoglobin oxygen ligation in the presence or absence of its allosteric ligands; see also *Henry et al., 1997*; *Milo et al., 2007*; *Rapp and Yifrach, 2017*; *Storz, 2016*). According to the MWC model, tetrameric Hb is assumed to be in equilibrium between the un-liganded low-affinity **T** (tense or deoxy) and the high-affinity **R** (relaxed or oxy) states, as described by the *L* allosteric constant (*L*=[**T**]/[**R**]). In the presence of oxygen, this equilibrium is shifted toward the **R** state in a manner determined by the ligation number and by the ratio of dissociation constants of oxygen from the two states, *c* (*c*=$K_R$/$K_T$). These aspects of the MWC model are quantitatively described by *Equation 2* (*Monod et al., 1965*):

$$\bar{Y}_{MWC} = \frac{\frac{[S]}{K_R}\left(1+\frac{[S]}{K_R}\right)^3 + L\frac{[S]}{K_T}\left(1+\frac{[S]}{K_T}\right)^3}{\left(1+\frac{[S]}{K_R}\right)^4 + L\left(1+\frac{[S]}{K_T}\right)^4} = \frac{a(1+a)^3 + Lca(1+ca)^3}{(1+a)^4 + L(1+ca)^4}$$

(2)

where α in the non-dimensional form of the MWC equation denotes substrate (S) concentration in $K_R$ units (α=[S]/$K_R$). In the classical MWC model, cooperativity in substrate binding (homotropic

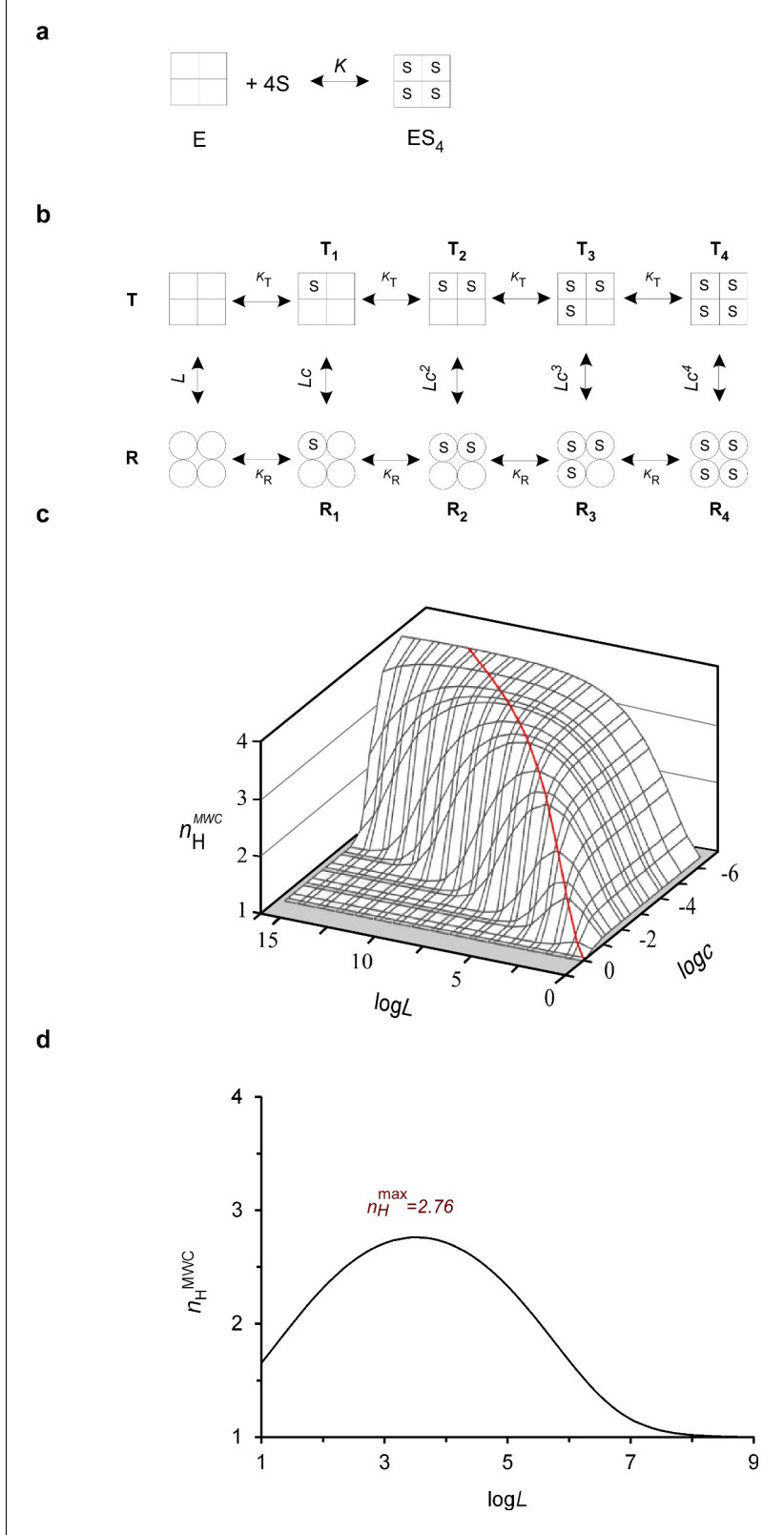

**Figure 1.** The biophysical Hill landscape of a tetrameric MWC protein. (a) Schematic representation of the all-or-none Hill binding mode by a tetrameric protein (**E**), with $K$ referring to the substrate (**S**) dissociation equilibrium constant. (b) Schematic representation of the MWC model applied to a tetrameric allosteric protein. Square and round symbols represent the tense (**T**) and relaxed (**R**) subunit conformations, respectively. $L$, $K_T$ and $K_R$ denote the
*Figure 1 continued on next page*

*Figure 1 continued*

T to R transition equilibrium constant in the absence of substrate and substrate affinity to the T and R conformations, respectively. The parameter *c* corresponds to the ratio of substrate affinity to the R and T conformations (=$K_R/K_T$). (**c**) The theoretical three-dimensional Hill landscape describing the dependence of $n_H^{MWC}$ at half-saturation on both the *L* and *c* parameters (calculated based on **Equation 5**). The red ridge line trajectory corresponds to parameter value pairs giving rise to maximum cooperativity. (**d**) Dependence of the Hill coefficient at half-saturation ($n_H^{MWC}$) on the allosteric constant *L*, as determined according to **Equation 5**. The curve was plotted assuming a *c* value of ~ 0.01, typical for the ratio of oxygen affinity to the R and T conformations of human Hb. The saddle point of the bell-shaped curve corresponds to a Hill value of ~ 2.8 (as observed for human Hb), fulfilling the $L = c^{-2}$ criterion giving rise to maximum cooperativity.

DOI: https://doi.org/10.7554/eLife.47640.003

interactions) is determined by both *L* and *c* (**Monod et al., 1965**). Heterotropic interactions, on the other hand, describing the effects of allosteric ligands on substrate binding, only affect the *L* parameter. Hb conformational equilibrium can thus be further shifted toward the **T** state upon binding of allosteric inhibitors. Such dependence is described by **Equation 3** assuming non-exclusive binding of the inhibitor (I) to both the **T** and **R** MWC conformations exhibiting $K_I^T$ and $K_I^R$ affinities, respectively (**Rubin and Changeux, 1966**):

$$L_{app} = L\left(\frac{\left(1 + [I]/K_I^T\right)}{\left(1 + [I]/K_I^R\right)}\right)^4 \tag{3}$$

Reliable values for the *L* and *c* MWC parameters of all evolutionary dataset Hb saturation curves and for the apparent *L* values of all physiological dataset human Hb saturation curves were recently reported (**Rapp and Yifrach, 2017**) and are given in **Supplementary files 1** and **2**, respectively. Values for $n_H$ and $p_{50}$ for the same dataset curves were previously reported (**Milo et al., 2007**; **Rapp and Yifrach, 2017**).

## The biophysical Hill landscape of a tetrameric MWC allosteric protein

To describe the driving force(s) that shape Hb affinity and cooperativity oxygen binding properties, one must first derive the exact relationship between $p_{50}$ and $n_H$ and the parameters of the MWC model. Accordingly, to bridge between the non-realistic Hill analysis and the MWC ligation pathway, we followed others (**Levitzki, 1978**; **Yifrach, 2004**) and employed **Equation 4** (**Wyman, 1964**) on the MWC model equation under conditions of half-saturation:

$$n_H = \frac{\partial log\left(\frac{\bar{Y}}{1 - \bar{Y}}\right)}{\partial(\log[S])} = \frac{[S]\partial(\bar{Y})/\partial([S])}{(\bar{Y})(1 - \bar{Y})} \tag{4}$$

Plugging the explicit mathematical expressions for $\bar{Y}_{MWC}$, $1 - \bar{Y}_{MWC}$ and $\partial(\bar{Y}_{MWC})/\partial([S])$ into **Equation 4** yields the explicit mathematical dependence of $n_H$ on the *L*, $K_R$ and $K_T$ biophysical parameters of the MWC model. Earlier studies (**Levitzki, 1978**; **Yifrach, 2004**) employed this analysis for assessing variations of a dimeric MWC protein to obtain analytic solutions for the dependence of $n_H$ on model parameters. Here, we solved for the tetrameric MWC protein case, yielding **Equation 5**:

$$n_H^{MWC} = \frac{\partial\left(\frac{\bar{Y}_{MWC}}{1 - \bar{Y}_{MWC}}\right)}{\partial(\log[S])} = \left(4[S]\left(\frac{\partial(\bar{Y}_{MWC})}{\partial([S])}\right)\right)|_{[S]=[S]_{0.5}} =$$

$$\left(4[S]\left(\frac{\frac{3[S]\left(1+\frac{[S]}{K_R}\right)^2}{K_R^2} + \frac{\left(1+\frac{[S]}{K_R}\right)^3}{K_R} + \frac{3L[S]\left(1+\frac{[S]}{K_T}\right)^2}{K_T^2} + \frac{L\left(1+\frac{[S]}{K_T}\right)^3}{K_T}}{\left(1+\frac{[S]}{K_R}\right)^4 + L\left(1+\frac{[S]}{K_T}\right)^4} - \frac{\left(\frac{4\left(1+\frac{[S]}{K_R}\right)^3}{K_R} + \frac{4L\left(1+\frac{[S]}{K_T}\right)^3}{K_T}\right)\left(\frac{[S]\left(1+\frac{[S]}{K_R}\right)^3}{K_R} + \frac{L[S]\left(1+\frac{[S]}{K_T}\right)^3}{K_T}\right)}{\left(\left(1+\frac{[S]}{K_R}\right)^4 + L\left(1+\frac{[S]}{K_T}\right)^4\right)^2}\right)\right)|_{[S]=[S]_{0.5}} \tag{5}$$

The expression for $[S]_{0.5}$ can be obtained by setting $\bar{Y}_{MWC}$ (**Equation 2**) to a value of 0.5. Given that [S] is raised to a power of up to 4, the exact analytic expression for $[S]_{0.5}(L, K_R, K_T)$ is not manually solvable and was obtained using Mathematica software (Wolfram Research Inc; see Materials

and methods). Substituting the solution for $[S]_{0.5}(L, K_R, K_T)$ in *Equation 5* then allows obtaining the explicit expression for $n_H^{MWC}$ in terms of the MWC model parameters. The long and complex expressions of both $[S]_{0.5}$ and $n_H^{MWC}$ can be found at the GitHub repository under the following address: https://github.com/OlgaRL/MWC_Parameters (*Rapp, 2019*; copy archived at https://github.com/elifesciences-publications/MWC_Parameters). Thus, for any $L$, $K_R$ and $K_T$ values, $n_H^{MWC}$ is unequivocally determined (*Equation 5*). A good approximation for $n_H$ (based on the $[S]_{0.5}$ expression of the $\bar{R}$ conformational state function; *Gruber et al., 2019*) that is over 98% accurate across most of the range of $L$ and $c$ values is presented in Appendix 2.

Plotting the three-dimensional surface of the Hill slope at half-saturation (now referred to as $n_H^{MWC}$) as a function of both $\log L$ and $\log c$ (based on *equation 5*) generates what we refer to as the theoretical 'biophysical Hill landscape' for a tetrameric MWC protein (*Figure 1c*). This landscape graphically describes the parameter space available for such a MWC protein and summarizes many of the known cooperativity-related aspects of the MWC model documented in classical papers (*Monod et al., 1965*; *Rubin and Changeux, 1966*; *Edelstein, 1975*; *Perutz, 1989*) and textbooks. In the Hill landscape, $n_H^{MWC}$ values between one to four are obtained as a function of the $L$ and $c$ values chosen. The $n_H^{MWC}$ surface presents a topology that resembles an upward-running ridge, with the maximum points obtained by introducing various $L$ and $c$ values (as indicated by the trajectory drawn in red). Such mapping is better appreciated in the bell-shaped dependence of $n_H^{MWC}$ on $\log L$ at a given $\log c$ value ($c = 0.01$, a value typical for human Hb), seen in a cross-section of such a surface (*Figure 1d*) (*Rubin and Changeux, 1966*; *Edelstein, 1971*). Inspecting the landscape cross-sections reveals that systematically higher maximum Hill points are obtained when higher $L$ and lower $c$ values are simultaneously chosen. Furthermore, as indicated by *Rubin and Changeux (1966)*, the different $L$ and $c$ value pairs underlying the maximum $n_H^{MWC}$ points must all fulfill a particular dependence expressed by $L = c^{-n/2}$, as obtained by setting $\partial n_H/\partial \log L$ equals zero. Another important aspect of the maximum Hill points is that they reside at the apex of rather wide hilltops. This can be seen in the bell-shaped cross-sections of $n_H^{MWC}$ as a function of $\log L$ at any particular $\log c$ value (*Figure 1c,d*), suggesting that around the maximum cooperativity points, only minor changes in $n_H$ are to be expected upon changes in $L$ brought about by binding of allosteric inhibitors or activators (*Equation 3*). This relative insensitivity of $n_H^{MWC}$ to changes in physiological conditions was referred to as a 'buffering of cooperativity' phenomenon by the MWC model founders (*Rubin and Changeux, 1966*; *Edelstein, 1971*). Using the biophysical Hill landscape described here, we can now examine where evolutionary and physiological considerations have affected Hb behavior.

## The characteristic $n_H$ value of hemoglobin is a maximum cooperativity point

Where along the three-dimensional $n_H^{MWC}$ landscape can one find the $(L, c, n_H)$ points of the different binding curves of the evolutionary and physiological Hb datasets? We first considered the 14 mammalian oxygen saturation curves comprising the evolutionary dataset. As can be seen in *Supplementary file 1*, the different mammal Hbs exhibit variations in both $L$ and $c$ values, thus giving rise to different Hill values (calculated based on the MWC model using *Equation 5*). These values resemble the observed 'Hill model'-derived coefficients (*Milo et al., 2007*), as indicated by the linear correlation between the observed and calculated Hill values reported by *Rapp and Yifrach (2017)* (*Figure 1* therein; see also *Supplementary file 1*). When mapped onto the theoretical Hill landscape, the calculated Hill values of all mammalian Hbs reside close to the maximum points of the bell-shaped contours of the three-dimensional Hill landscape (*Figure 2a*). Furthermore, the logarithms of the values for $L$ and $c$ of the different Hbs are linearly correlated (*Figure 2b*) with a slope value of $-2.7$ ($\pm 0.3$), close to a value of $-2$ that would be expected for proteins that evolved to present maximum cooperativity in ligand binding ($L = c^{-n/2}$). The observation that the $\log L$-$\log c$ correlation crosses very close to the $(0,0)$ axis origin point further supports this assertion. Given that $L$ and $c$ are independent parameters of the MWC model, the logarithms of their values for a given protein set are not expected to be linearly-dependent. This is, in particular, true if one considers that these values were derived using reliable data fitting strategies that prevent potential parameter dependence artifacts (*Rapp and Yifrach, 2017*) and that changes in either $c$ or $L$ (or in both) were reported for native or mutant Hb variants over many years of Hb research. We thus suggest that the observed

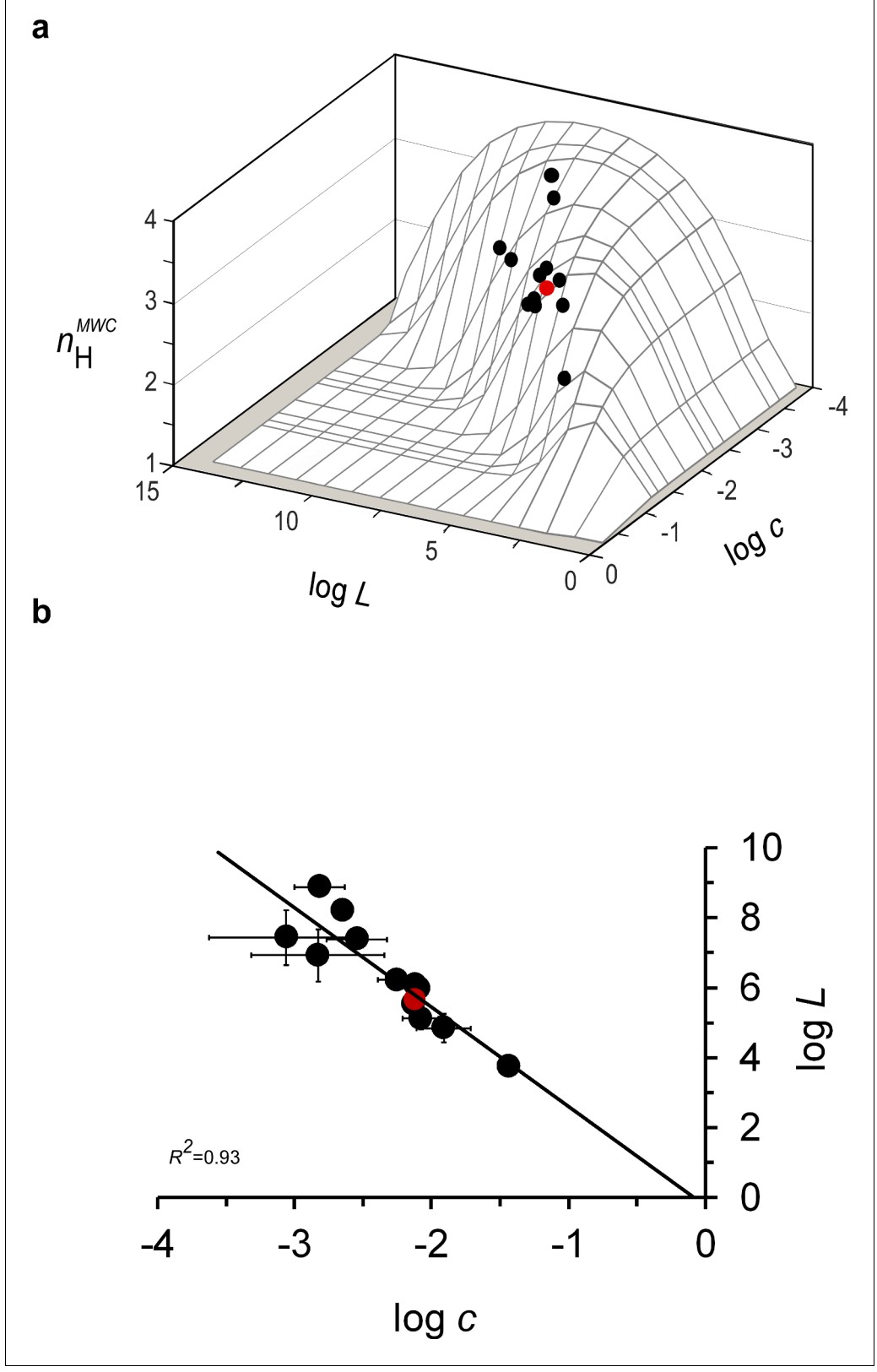

**Figure 2.** Mammalian hemoglobins exhibit maximum cooperativity $n_H$ values. (a) $n_H^{MWC}$ values of the different mammal Hbs from the evolutionary dataset, calculated based on derived values for the $L$, $K_T$ and $K_R$ elementary MWC parameters, reside close to the ridge line of the three-dimensional biophysical Hill landscape. (b)
*Figure 2 continued on next page*

*Figure 2 continued*

Correlation plot relating the logarithms of the relative affinity (*c*) and conformational ratio (*L*) of the **T** and **R** quaternary MWC states of different mammalian Hbs comprising the evolutionary dataset (*Supplementary file 1*). The solid curve corresponds to a linear regression with a $R^2$ correlation coefficient of 0.93, a slope of -2.7 (± 0.3) and an intercept at log$L$ of -0.2, very close to the 0 value expected based on the $Lc^2$=1 criterion. In both panels, the red data point corresponds to the values for human Hb.

DOI: https://doi.org/10.7554/eLife.47640.004

linearity between mammalian log$L$ and log$c$ values does not imply causation or is an artifact but rather points to co-evolution of the two MWC parameters so as to shape Hb cooperativity close to the maximum value. Deviation between the observed and expected slope values of the log$L$-log$c$ correlation can be explained by differences in the techniques and experimental protocols used to evaluate the Hb saturation data from the different organisms and inaccuracies in data collection and analysis (for detailed discussion, see *Milo et al., 2007*). Nevertheless, even with such limitations, the results derived from the evolutionary dataset support the hypothesis that changes in mammalian Hb binding and conformation have been tuned during evolution, giving rise to close to maximum cooperativity in oxygen binding, albeit yielding a different maximum $n_H$ value in each case. To further examine whether this indeed is the case, we considered the human Hb physiological dataset.

As highlighted above, a buffering of cooperativity regime is observed around a maximum cooperativity Hill point. As such, minor changes in $n_H$ are to be expected upon changes in $L$ brought about by the binding of allosteric inhibitors or activators. Reliable estimates for the apparent $L$ values of all of 26 different concentration-related saturation curves of the $H^+$, $CO_2$ and 2,3-BPG physiological datasets were recently obtained (*Rapp and Yifrach, 2017*), assuming a constant *c* value in each case, as predicted by the MWC model (*Supplementary file 2*). The different $L_{app}$ values obtained for each effector dataset saturation curve scales with effector concentration in a manner predicted by the MWC model (*Rapp and Yifrach, 2017*; Appendix 1), confirming the reliability of the values derived.

For the physiological Hb dataset, a cross-section of the biophysical Hill landscape at the characteristic *c* value of human Hb should be considered. We thus examined where along the bell-shaped human Hb $n_H$-log$L$ curve do the characteristic pairs of log$L$-$n_H$ values of the 26 physiological dataset curves reside. As can be seen in *Figure 3a*, log$L$-$n_H$ data points of the physiological dataset (black symbols) cluster around the maximum point of the bell-shaped curve. This pattern is in sharp contrast to the scattered pattern of the pairs of log$L$-$n_H$ values for Hb mutants reported in the literature (gray circles) (*Baldwin, 1976*) and assumed by *Baldwin (1976)* (and later by *Fersht, 1985*) to have the same *c* value. Significant deviations from the maximum cooperativity value of wild-type human Hb were also observed for naturally occurring Hb mutants, such as the Kansas and Chesapeake variants (*Edelstein, 1971*). A clustering pattern similar to that shown for human Hb is also seen for bovine and dog Hbs, despite being based on the limited physiological datasets available for these species (*Figure 3b*) (*Jensen, 2004*; *Breepoel et al., 1981*).

The clustering of all log$L$-$n_H$ experimental data points of the $H^+$, $CO_2$ and 2,3-BPG datasets around the maximum point reveals that the Hill value for human Hb is relatively insensitive to changes in physiological conditions. Such $n_H$-insensitive behavior was noted for the pH-related Bohr effect of Hb (*Wyman, 1964*; *Di Cera et al., 1988*) and later extended to other allosteric effectors of Hb (*Milo et al., 2007*). Our results showing the invariance of $n_H$ with respect to log$L$ thus provide a mechanistic explanation, based on the MWC model parameters, for the 'buffering of cooperativity' property of Hb. Such behavior should only appear if human Hb had evolved to present maximum cooperativity in oxygen binding (*Figure 1c,d*). As site-directed point mutants of Hb are not subjected to the selection pressure that leads to maximum cooperativity, the values of log$L$-$n_H$ pairs for such mutants fall off the extremum point of the log$L$-$n_H$ cross-section (*Baldwin, 1976*).

## The maximum Hill point of a MWC-type protein allows for gross- and fine-tuning of substrate unbinding sensitivity

We next explored the potential evolutionary and physiological advantages of setting Hb cooperativity around the maximum $n_H$ point. We asked the following questions: Does the value of the maximum Hill point determine the sensitivity of Hb to tissue oxygen unloading? Does the value of the

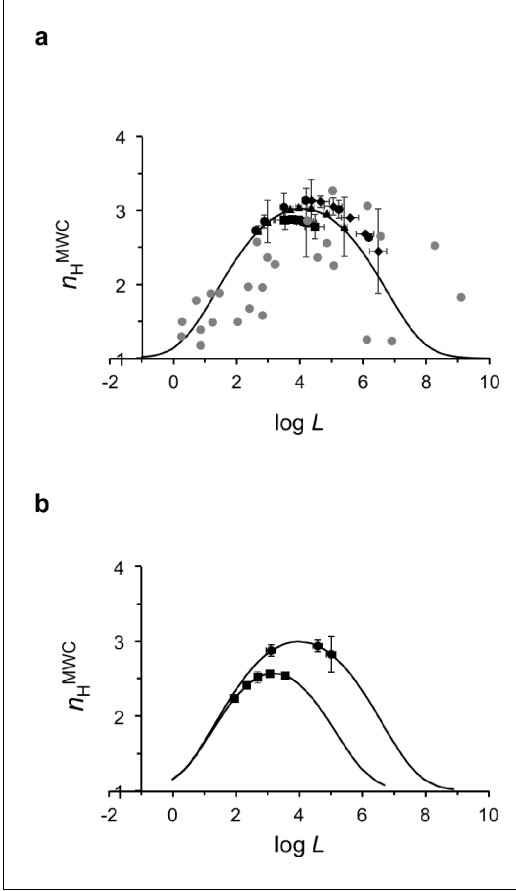

**Figure 3.** A buffering of cooperativity physiological regime is observed for human hemoglobin around its maximum cooperativity value. (a) Mapping of the Hill coefficient and the logarithm of the apparent allosteric constant (*L*) values for all human oxygen saturation curves of the pH, $CO_2$ and BPG physiological datasets (*Supplementary file 2*) onto the theoretical $n_H$-*logL* curve of human Hb. The $n_H$ values for all physiological dataset curves were calculated using the derived apparent *L* value for each curve and assuming a common *c* value. The theoretical $n_H$-*logL* curve was calculated assuming a *c* value of 0.007 (±0.001), corresponding to the averaged *c* value of all human physiological datasets (*Supplementary file 2*). Black circles, squares and diamonds correspond to pH, $CO_2$ and BPG data points, respectively. Gray symbols represent pairs of *logL*-$n_H$ values for Hb mutantS reported in the literature and assuming no change in *c* upon mutation (*Baldwin, 1976*). (b) Dependence of the Hill coefficient at half-saturation on the apparent allosteric constant for all oxygen saturation curves of the dog and bovine physiological datasets (squares and circles, respectively). For each species, the appropriate *c* value (*Supplementary file 1*) was used to calculate the theoretical curve, onto which the actual *logL*-$n_H$ data points were mapped.

DOI: https://doi.org/10.7554/eLife.47640.005

maximum $n_H$ point affect the dynamic range of the buffering of cooperativity phenomenon? How is Hb protein unbinding sensitivity changes within the buffering of cooperativity range? To answer these questions, we turned to *Equation 2* and the biophysical Hill landscape (*Figure 1c*).

Using the MWC equation, we generated ligand saturation binding curves exhibiting increasing maximum Hill cooperativity values (*Figure 4a*). This was achieved by appropriate selection of *L* and *c* values (indicated in *Table 1*), all fulfilling the $L=c^{-2}$ parameter criterion. To allow direct comparison, the saturation curves were plotted as a function of the scaled concentration $\alpha$ (=[S]/$K_R$), as is commonly done (*Monod et al., 1965*; *Rubin and Changeux, 1966*). Under such conditions, the affinity and maximum Hill point are correlated (*Figure 4a*). The higher the value of the maximum Hill point of the appropriate binding curve, the more it is displaced to the right along the scaled concentration axis. This is reflected in *Figure 4b*, showing monotonic correlation between the maximum Hill value and $\log\alpha_{1/2}$ ($\alpha_{1/2}$=[S]$_{1/2}$/$K_R$). It thus seems that under the maximum cooperativity parameter condition, a tradeoff exist between the cooperativity and affinity properties of a MWC-type protein. We next examined if and how this tradeoff affects the sensitivity of the protein for substrate unbinding upon chemical potential change. For each of the curves presented in *Figure 4a*, we considered the chemical potential difference ($\Delta\log\alpha$) needed to achieve a constant reduction in the fractional saturation from 100 to 40%, as typically observed for mammalian Hb saturation under physiological lung and tissue oxygen pressures (100 and 40 mmHg, respectively). The $\Delta\bar{Y}_{100 \rightarrow 40\%}$/$\Delta\log\alpha$ parameter depends on both the affinity and cooperativity attributes of the binding curve and reflects the sensitivity of the MWC protein to substrate unloading. The narrower the difference in chemical potential that results in a 60% reduction in fractional saturation, the higher is the sensitivity of the protein to substrate unbinding. As can be seen in *Figure 4c* (filled black symbols), the value for this parameter increases monotonically in a power law manner with increases in the value of the maximum Hill point. Similar results were obtained when other $\Delta\bar{Y}$ ranges were considered (e.g. $\Delta\bar{Y}_{100 \rightarrow 20\%}$; not shown). Higher maximum Hill points thus ensure higher sensitivity for substrate unbinding by the MWC protein.

We next addressed the dynamic range implicated in physiology adaptation. The maximum $n_H$ points of the different curves shown in *Figure 4a*

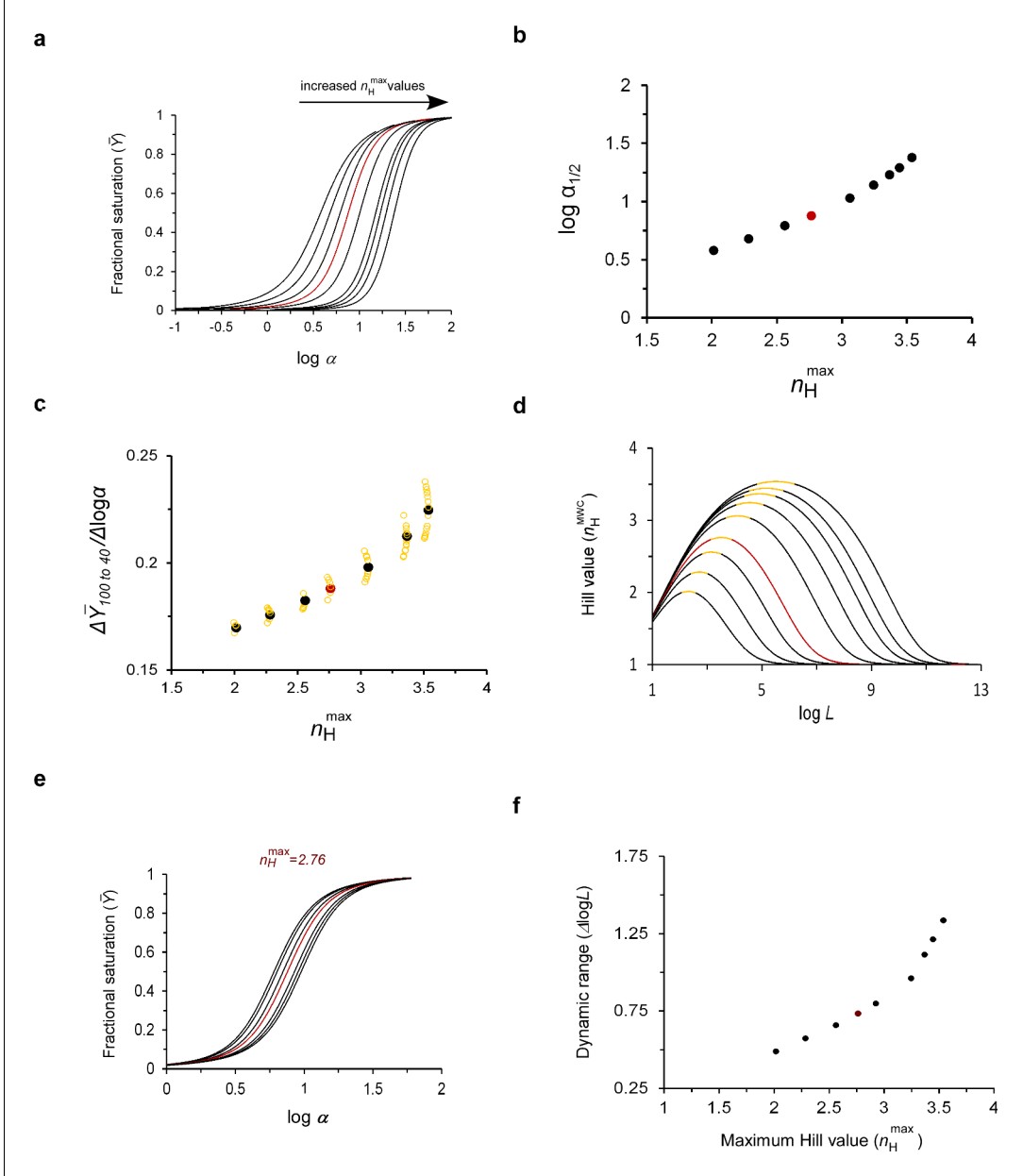

**Figure 4.** The maximum cooperativity point determines MWC protein substrate unbinding sensitivity and the dynamic range of physiological adaptation. (a) Fractional saturation binding curves of a MWC protein as a function of the logarithm of the scaled concentration $\alpha$, with parameter values that give rise to systematically higher maximum cooperativity $n_H$ values. The pairs of $L$ and $c$ values for all curves obey the maximum cooperativity criterion and are listed in *Table 1*. (b) Dependence of the logarithm of the scaled affinity ($\log\alpha_{1/2}$) of the curves presented in (a) on the maximum cooperativity Hill value ($n_H^{max}$). (c) Dependence of the substrate unbinding sensitivity ($\Delta\bar{Y}_{100 \to 40\%}/\Delta\log\alpha$) of a tetrameric MWC protein on $n_H^{max}$ (filled black symbols). The array of open yellow symbols decorating each maximum cooperativity point corresponds to changes in the Y axis values due to changes in $L$ that reside within the correspondingly encoded buffering of cooperativity regime (see text) (d) Dependence of $n_H^{MWC}$ on the logarithm of the allosteric constant $L$ around the different maximum cooperativity points indicated in (a) and *Table 1* (and assuming their different $c$ values). For each curve, the portion highlighted in yellow corresponds to the $\log L$ range fulfilling the 'buffering of cooperativity' regime, that is the $L$ range around the maximum Hill point that gives rise to fractional saturation curves all exhibiting less than 1% change in the maximum Hill coefficient value. Such a physiological dataset is observed for the maximum point of ~2.8, characteristic of human Hb in panel (e) (calculated using its appropriate $c$ value; see *Supplementary file 1*). Similar data for different maximum cooperativity values are presented in *Figure 4—figure supplement 1*. In each physiological dataset, the curve indicated in red corresponds to the maximum cooperativity curve (f) Dependence of the broadness of the 'buffering of cooperativity' regime (the dynamic range), as defined above, on the maximum cooperativity Hill value. In all figure panels, the curves indicated in red were calculated using the experimental values for human Hb reported in the literature. The red data points in several of the panels represent approximated values of human Hb.

*Figure 4 continued on next page*

*Figure 4 continued*

DOI: https://doi.org/10.7554/eLife.47640.006

The following figure supplement is available for figure 4:

**Figure supplement 1.** Simulated MWC-based physiological saturation data around different maximum cooperativity $n_H$ values.

DOI: https://doi.org/10.7554/eLife.47640.007

reside precisely at the apex of the appropriate bell-shaped $n_H$-$\log L$ curve, calculated using the appropriate *c* value (*Figure 4d* and *Table 1*). As mentioned earlier, around this maximum $n_H$ point cooperativity is invariant to changes in *L* brought about by binding of allosteric modulators. To assess the broadness of the buffering of cooperativity regime around each maximum cooperativity point, we next generated a set of physiological dataset curves by systematically changing the value of *L* by several orders of magnitude, such that the resulting saturation curves exhibited essentially the same Hill value for ligand binding as did the original maximum Hill value (less than a 1% change). This ensured that our analysis remained within the 'buffering of cooperativity' region, characteristic of each maximum Hill value (see the yellow-marked $\log L$ range highlighted in each of the curves in *Figure 4d*). One such physiological dataset generated around the maximum point of human Hb ($n_H = \sim 2.8$) is shown in *Figure 4e*. For curves based on other maximum cooperativity points, see *Figure 4—figure supplement 1*. For each of the different datasets, we calculated the $\log L$ range between the two extreme binding curves and defined this value ($\Delta \log L$) as the broadness of the buffering of cooperativity regime, or the dynamic range for physiological adaptation where only affinity ($p_{50}$) matters. Plotting the dependence of $\Delta \log L$ on the maximum cooperativity Hill value (*Figure 4f*) reveals that higher maximum Hill values give rise to broader *L* ranges, where 'buffering of cooperativity' is observed. The maximum Hill point thus further controls the dynamic range of physiological adaptation.

Lastly, we considered how protein sensitivity to substrate unbinding, as defined above, varied within the yellow marked buffering of cooperativity regime (*Figure 4d*). Thus, for each set of fractional saturation curves calculated around the different maximum cooperativity points and residing within this regime (*Figure 4e* and *Figure 4—figure supplement 1*), we calculated $\Delta \bar{Y}_{100 \to 40\%}/\Delta \log \alpha$. Variations in this parameter are plotted as open yellow symbols around each of the maximum cooperativity points presented in *Figure 4c*, with the lower and upper data points, respectively, corresponding to the lower and upper $\log L$ values defining the boundaries of this range. As can be seen, within this range, changes of up to 6% are observed in the value of protein unbinding sensitivity in

**Table 1.** Values for the *L* and c MWC parameters giving rise to systematically higher maximum cooperativity Hill values[a]

| c | L | Maximum Hill value ($n_H^{max}$) |
|---|---|---|
| 0.07071 | 200 | 2.02 |
| 0.04472 | 500 | 2.28 |
| 0.02582 | 1500 | 2.56 |
| **0.01741** | **3300** | **2.76** |
| 0.01222 | 6700 | 3.06 |
| 0.00524 | 36400 | 3.24 |
| 0.0035 | 81800 | 3.37 |
| 0.00262 | 145400 | 3.44 |
| 0.00175 | 327200 | 3.54 |

[a]Values for the *c* and *L* parameters giving rise to the maximum Hill slopes (indicated in the right-most column), characteristic of the binding saturation curves analyzed in *Figure 4a*. All parameter sets meet the $Lc^2 = 1$ criterion. The parameter set giving rise to the maximum cooperativity of human Hb, as reported in the literature and in *Supplementary file 1*, is indicated in bold. The different c values indicated were obtained by varying the $K_R$ affinity parameter while assuming a constant $K_T$ value of 143 (see *Supplementary file 1*).

DOI: https://doi.org/10.7554/eLife.47640.008

response to changes in physiological conditions brought about by changes in $L$ alone yet reflected in changes in $p_{50}$ alone.

## The maximum cooperativity point of hemoglobin is related to organismal fitness

The fact that different mammalian Hbs exhibit different maximum $n_H$ values (*Figure 2a*), combined with the observation made above whereby systematically higher maximum Hill value gives rise to higher sensitivities to substrate unloading by MWC proteins (*Figure 4c*), led us to hypothesize that the maximum cooperativity point of Hb has been tuned during evolution to satisfy the particular physiological requirements of different tissues for oxygen supply. We thus returned to the datasets comparing oxygen saturation of different mammalian Hbs (all measured using similar methodology) that revealed the linear correlation between animal metabolic rate and log$p_{50}$ (*Schmidt-Neilsen and Larimer, 1958*; *Schmidt-Nielsen, 1970*). We are aware that these data may not necessarily reflect functional effects on a single Hb form. Still, in cases where adult mammalian Hb expresses more than one variant, both usually present similar sequences and functional properties (*Storz, 2016*). Rather than taking the affinity-centric approach employed in the earlier studies, we instead adopted a complementary cooperativity-centric mindset and re-fitted all of the saturation curves to the Hill equation to derive both $n_H$ and $p_{50}$ values for Hb of each animal (*Figure 5a* and *Table 2*). The derived $n_H$ values, although obtained under the slightly different physiological conditions typical of the blood of each animal, nevertheless correspond to maximum cooperativity values, as $n_H$ is relatively invariant to physiological changes around the maximum point (*Figures 2* and *3*). This assertion is further strengthened by the results shown in *Figure 5b* describing the linear relation between affinity (log$p_{50}$) and cooperativity ($n_H$), as expected for parameter values underlying maximum cooperativity (*Figure 4b*). Finally, plotting either the affinity ($\sim$log($1/p_{50}$)) or cooperativity properties of Hb as a function of the logarithm of metabolic rate of an animal yielded linear correlations in both cases (*Figure 5c*). Specifically, the greater an animal's metabolic rate, the higher was the observed maximum Hill cooperativity value of oxygen binding by Hb. Furthermore, as previously shown (*Schmidt-Neilsen and Larimer, 1958*; *Schmidt-Nielsen, 1970*), an inverse linear correlation was found between the metabolic rate and Hb affinity. These empirical observations directly relate the maximum cooperativity and affinity properties of Hb to an organismal-level trait that is related to fitness (*Dykhuizen et al., 1987*; *Kacser and Burns, 1981*; *Bershtein et al., 2017*).

## Discussion

In the current study, we emphasized the need to consider the property of Hb cooperativity ($n_H$), along with $p_{50}$, in addressing molecular adaptation of Hb in mammals. We formulated the explicit mathematical dependences of the $n_H$ and $p_{50}$ macroscopic Hb parameters on the elementary microscopic parameters of the MWC model, thus providing a quantitative framework to understand the mechanism underlying the adaptive behavior of these two parameters with respect to tissue oxygen delivery. Indeed, relying on the biophysical Hill landscape, we were able to rationalize mammalian Hb $n_H$ and $p_{50}$ diversity among large and small body-sized animals. The biophysical Hill landscape mapping strategy may further serve to define the mechanism underlying other molecular adaptation modes involving Hb of organisms living at high altitudes or presenting burrowing or diving lifestyles, for example.

### Mammalian Hb diversity resides on the ridge of maximum cooperativity

Our findings revealed that mammalian Hb cooperativity values have been tuned during evolution to reside on a ridge of the theoretical biophysical Hill cooperativity landscape describing a MWC protein (*Figures 2* and *3*). These ridge-located Hill values all correspond to maximum cooperativity extremum points and were achieved by tuning of the relative affinity ($c$) and conformational ratio ($L$) of the **T** and **R** quaternary states of the protein (*Figure 2*). These maximum points are thus unique points along the Hill cooperativity scale continuum. There is no a priori reason to expect that physiological and evolutionary diversity in Hb would be reflected by this low dimensional ridge feature of the Hill landscape. As can be seen in the three dimensional Hill landscape, many combinations of $L$ and $c$ values can give rise to the same Hill value. Yet, only one of these combinations obeys the $L = c^{-2}$ maximum cooperativity criterion. The observed variations in maximum cooperativity Hill

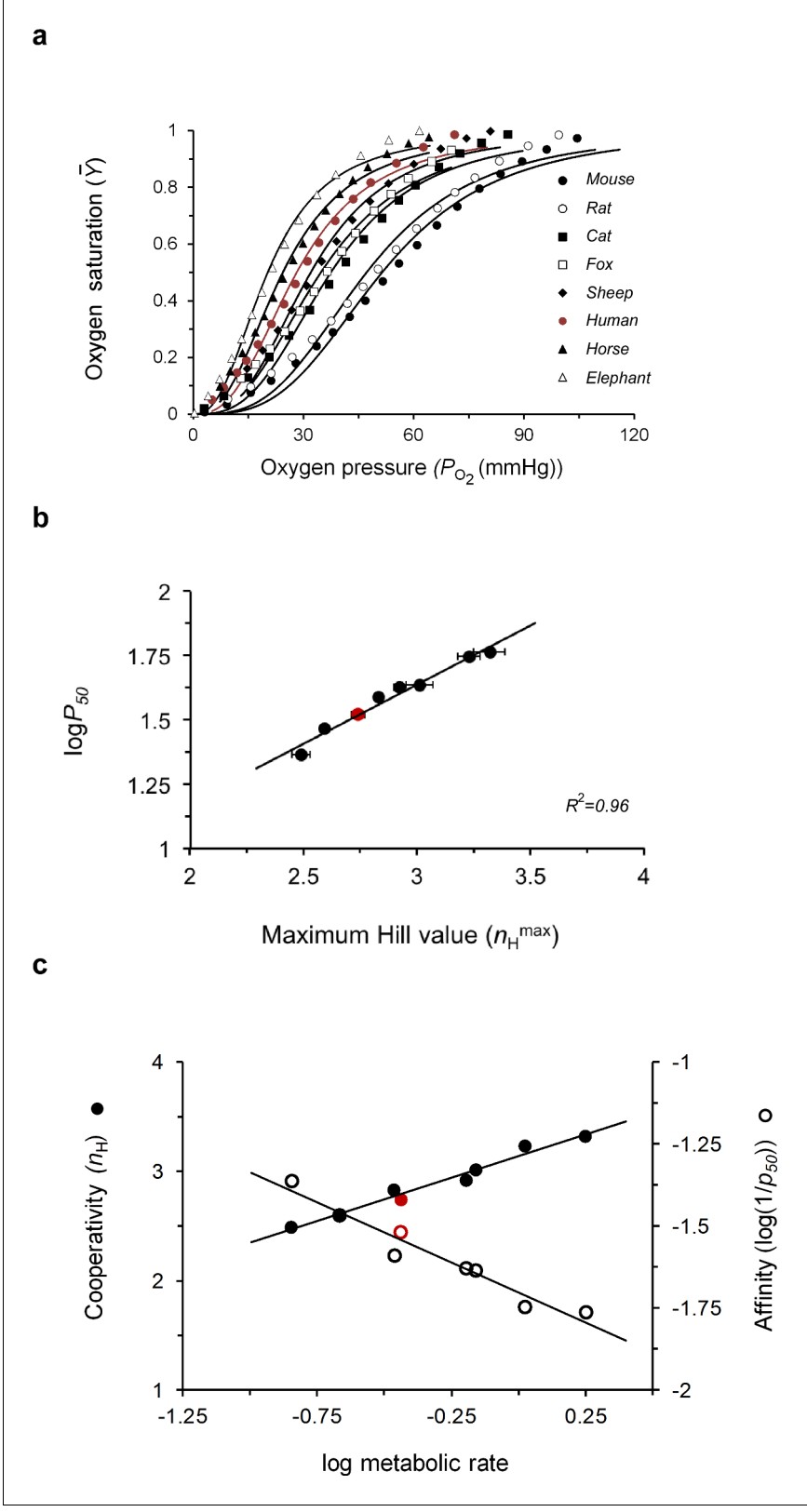

**Figure 5.** Evolutionary tuning of the maximum cooperativity and affinity points of mammalian hemoglobins underlies physiological adaptation. (a) Dependence of the fractional oxygen saturation of animal blood sample Hbs on partial oxygen pressure, as reported by *Schmidt-Neilsen and Larimer (1958)*, *Schmidt-Nielsen (1970)*. Solid curves represent data fitting to the Hill equation (*Equation 1*). $p_{50}$ and $n_H$ values for oxygen binding to the

*Figure 5 continued*

different mammalian Hb datasets are reported in *Table 2*. (b) Correlation plot relating the maximum $n_H$ and
$logp_{50}$ values of the curves shown in a. (c) Dependence of the values for both cooperativity (filled symbols) and
affinity (open symbols) properties of different animal Hbs on normalized metabolic rates in that animal (relative to
body weight). A linear trend is observed in both cases, with $R^2$ values of 0.97 and 0.9 for the cooperativity and
affinity correlations, respectively. Values for animal metabolic rates were taken from *Schmidt-Neilsen and Larimer
(1958)*. The red curve or data points presented in panels **5a-c** correspond to the values for human Hb.

DOI: https://doi.org/10.7554/eLife.47640.009

values among mammalian Hb proteins thus reflect co-evolution of binding and conformational
changes of Hb. Such tuning gives rise to a buffering of cooperativity effect, suggesting that the
degree of cooperativity is robust to changes in physiological conditions brought about by tuning $L$
alone (*Figure 3*). Our results thus reveal the evolutionary strategy underlying the observed variations
in Hb behavior.

## Evolutionary and physiological adaptations of hemoglobin serve to optimize tissue oxygen delivery

The functional advantages of the balance between the relative affinity and conformational ratio of
Hb quaternary states underlying the maximum Hill point become clear when considering that both
the affinity and cooperativity parameters of Hb saturation curves affect tissue oxygen unloading.
First, the $c$-$L$ tradeoff gives rise to the highest possible Hill value for any given $c$ value, thereby ensur-
ing maximal oxygen unloading upon Hb transfer from the lungs to other tissues (*Figure 1cd*). Either
very high or very low $L$ values would give rise to hyperbolic oxygen saturation curves with an $n_H$
value of one (*Figure 1c*), which would ensure poor and unregulated tissue oxygen unloading by Hb.
The same is true when $c$ is assigned the value of one. Second, tuning the Hb maximum Hill point to
higher values further leads to greater sensitivity to tissue oxygen unloading (*Figure 4c* filled, sym-
bols). Third, parameter adjustments that lead to higher maximum $n_H$ values also give rise to reduced
Hb oxygen affinity ($p_{50}$) (*Figures 4b* and *5b*), thus reinforcing oxygen unloading upon Hb transfer
from the lungs to other tissues. Indeed, the affinity-cooperativity tradeoff typical of the ridge of max-
imum cooperativity suggests that cooperativity can compensate for changes in affinity, with respect
to tissue oxygen unloading, as usually described in any *Biochemistry* textbook. Fourth, the balanced
parameter values give rise to the 'buffering of cooperativity' regime, thus ensuring that the $n_H$-
related oxygen unloading effect is not compromised in response to physiological changes brought
about by variations in $L$ and reflected as changes in $p_{50}$. Indeed, under such a regime, relatively
minor changes in Hb sensitivity for oxygen release are expected in response to changes in allosteric
effector binding (*Figure 4c*, open symbols). Finally, the co-varied $c$ and $L$ values underlying

**Table 2.** Values for the affinity and cooperativity parameters of animal blood hemoglobin binding
curves[a]

| Animal | $P_{50}$ (mmHg) | $n_H$ |
|---|---|---|
| *Elephant* | 23.03 ± 0.32 | 2.49 ± 0.04 |
| *Horse* | 29.26 ± 0.16 | 2.59 ± 0.01 |
| *Human* | **33.02 ± 0.29** | **2.74 ± 0.03** |
| *Sheep* | 38.82 ± 0.11 | 2.83 ± 0.01 |
| *Fox* | 42.36 ± 0.22 | 2.92 ± 0.02 |
| *Cat* | 43.22 ± 0.62 | 3.01 ± 0.06 |
| *Rat* | 55.63 ± 0.57 | 3.23 ± 0.05 |
| *Mouse* | 57.86 ± 0.76 | 3.32 ± 0.07 |

[a]Values for the point of half-saturation ($P_{50}$) and the Hill coefficient at half-saturation ($n_H$) were derived by fitting
blood sample animal Hb oxygen saturation curves (*Schmidt-Neilsen and Larimer, 1958*; *Schmidt-Nielsen, 1970*) to
the Hill equation.

DOI: https://doi.org/10.7554/eLife.47640.010

maximum cooperativity allow for control of the broadness of the 'buffering of cooperativity' regime (*Figure 4d,f*).

Taken together, these points support the assertion that mammalian Hb $n_H$ and $p_{50}$ values reflect evolutionary and physiological adaptations that serve to optimize the tradeoff between $O_2$ loading in pulmonary capillaries and $O_2$ unloading in tissue capillaries, so as to optimize tissue oxygen delivery. Based on experimental (*Figures 2* and *3*) and theoretical (*Figure 4*) results, we suggest that evolutionary (*i.e.*, across species) variations allow for gross changes in Hb oxygen unloading sensitivity. Such changes are controlled by the maximum cooperativity and affinity values of Hb, in turn determined by changes in both $L$ and $c$ (*Figure 4c*, filled symbols). Fine-tuning changes in Hb oxygen unloading sensitivity may further be brought about by physiological variations within the buffering of cooperativity regime, this time, however, reflected in changes in $L$ and $p_{50}$ alone (*Figure 4c*, open symbols).

## The biophysical Hill landscape as a low dimensional biophysical fitness landscape

The linear dependence between the metabolic rate of an animal and the maximum Hill value for Hb (*Figure 5c*), combined with the results presented in *Figure 4c*, indicates that $n_H$, like $p_{50}$, carries adaptive value with respect to mammalian Hb tissue oxygen unloading. As such, small body-sized animals presenting high metabolic rates require high oxygen supplies that can be more efficiently delivered by Hb presenting higher maximum Hill values. Considering that an animal's metabolic rate is a trait related to organism fitness, we suggest that both the affinity and cooperativity traits of Hb represent low dimensional components of the fitness landscape, independent of other components. In this respect, the biophysical $n_H$ landscape of Hb can be regarded as a biophysical fitness landscape that serves to bridge the genotype-phenotype gap in assessing the relation between sequence variations in Hb (that affect its affinity and cooperativity properties) and their ensuing fitness effects (*Bershtein et al., 2017*). Evolutionary tuning of mammalian Hb maximum cooperativity and affinity points, achieved by co-varying the relative affinity and conformational ratio of its quaternary states, thus regulates tissue oxygen delivery by Hb and serves as one mean to meet the distinct needs of different animal tissues for oxygen (*Figures 4* and *5*). Clearly, other mechanisms related to cardiac output and to convective and diffusive steps in the oxygen transport pathway are as well contribute to modulation of tissue oxygen transport (reviewed in *Schmidt-Nielsen, 1975*). For example, it has been shown that the capillary density of a mammal is also inversely correlated to body size (*Schmidt-Nielsen, 1975*).

## How general are our findings regarding maximum cooperativity in ligand binding?

The buffering of cooperativity regime in Hb reflects the case where the fractional saturation curves of the protein seem parallel in shape and are only displaced along the substrate concentration axis in response to changes in effector ligand concentrations (see *Figure 4e*). As revealed by *Olsman and Goentoro (2016)*, it is exactly this behavior the qualifies a MWC protein as a logarithmic sensor, reflecting the capacity of the protein to respond identically over a broad range of changes in chemical potential. Such behavior is in contrast to the case where the protein sensor responds to the absolute value of the signal intensity. Upon surveying the literature, *Olsman and Goentoro (2016)* found that many MWC allosteric proteins, including phosphofructokinase, Hb, cyclic nucleotide-gated ion channels, Tar receptors, G-protein coupled receptors and EGF receptors exhibit a buffering of cooperativity or logarithmic sensing regime in response to changes in the concentration (or magnitude) of the modulatory allosteric signal. The dynamic range for this behavior can vary from 10-fold (Hb) to 10,000-fold (GPCR). These findings suggest that all these MWC proteins evolved to present maximum cooperativity values in ligand binding and operate under the $L = c^{-n/2}$ parameter regime that gives rise to the buffering of cooperativity phenomenon. However, other than Hb, the significance of this finding to organismal fitness is known only in the case of a single protein (*Keymer et al., 2006*). Realizing the physiological advantage(s) of attaining the maximum cooperativity point of any protein and its relation to organismal fitness, is, of course, case-specific.

## The relation between physiological and evolutionary adaptations in shaping a protein molecular property

Finally, the roadmap that underlies the 'biophysical Hill landscape'-mapping strategy is general and can be applied to reveal the relation between evolutionary (i.e. sequence) and physiological (e.g. changes in metabolite concentrations) variations in shaping any molecular property of a protein. Although operating on different time scales and through different mechanisms, these traits are linked through the biophysical model describing the function of the protein. Elucidating this linkage requires extensive physiological and evolutionary datasets of the protein and that the steady-state biophysical model describing the function of the protein is known so that the explicit mathematical dependence of the molecular property under study on biophysical model parameters can be elucidated. This allows for constructing a 'biophysical molecular property landscape' onto which experimentally derived values of the macroscopic and microscopic parameters of all physiological and evolutionary dataset measurements can be mapped. Rationalizing the pattern of such mapping in terms of the relation between model parameters reflecting particular protein processes, such as catalysis, binding or conformational changes, might then reveal the driving forces that shape the molecular property under study.

# Materials and methods

## Datasets

The functional data analyzed here considered the evolutionary and physiology Hb datasets respectively complied in *Milo et al. (2007)* and *Rapp and Yifrach (2017)*. The evolutionary dataset comprised oxygen saturation curves of Hb from 14 different mammal samples obtained under similar physiological conditions. Using a three-equation system strategy (*Rapp and Yifrach, 2017*), the values for the $L$ and $c$ MWC allosteric parameters of all dataset curves were recently reported (*Supplementary file 1*). The physiological dataset comprised four independent human Hb sub-datasets, obtained from three different labs, with two reporting pH modulation, one considering $CO_2$ modulation and the last addressing 2,3-BPG modulation, as discussed in the meta analyses by *Rapp and Yifrach (2017)*. These sub-datasets each include between 5 and 7 oxygen saturation curves obtained at different effector concentrations. Using global fitting analysis, we recently reported reliable estimates for the apparent $L$ values for the different concentration-related physiological datasets curves (see *Rapp and Yifrach, 2017* and *Supplementary file 2*).

## The biophysical Hill landscape

Cooperativity in oxygen binding by Hb was assessed using the Hill or MWC models, as quantitatively described by *Equation 1 and 2*. The dependence of $n_H$ at half-saturation ($n_H^{MWC}$) on the MWC elementary parameters was obtained by applying *Equation 4* to the MWC equation, assuming $\bar{Y}_{MWC}$ equals ½. Explicit solving for $[S]_{1/2}$ in terms of $K_R$, $K_T$ and $L$ was obtained using the Mathematica 11.2 software package (Wolfram research Inc). Substituting this expression in *Equation 5* using the same software yields the explicit expression for $n_H^{MWC}$ (at half-saturation) in terms of the MWC model parameters. It should be noted that $n_H^{MWC}$ (at half-saturation) is an approximate measure of $n_{H,\,max}^{MWC}$, the more natural Hill value for specifying homotropic cooperativity in the framework of the MWC model (Rubin and Changuex, 1966). In our analysis, we used $n_H^{MWC}$ rather than the $n_{H,\,max}^{MWC}$ as the former is the actual parameter that is usually evaluated when fitting experimental data to the Hill equation (*Equation 1*). As for $n_{H,\,max}^{MWC}$, the expression for $n_H^{MWC}$ at half-saturation also exhibits a bell-shaped dependence on $L$, thereby also describing the characteristic 'buffering of cooperativity' effect. Furthermore, $n_{H,\,max}^{MWC}$ is equal to $n_H^{MWC}$ exactly at the maximum cooperativity point (*Rubin and Changeux, 1966*). It should be emphasized that the $n_H^{MWC}$ expression derived here relates to cooperativity in ligand binding (as it is calculated based on $\bar{Y}$) and is different than the expressions for $n_H$ derived in other studies based on the active state function ($\bar{R}$ in the case of an enzyme or $\bar{P}_{open}$ in case of an ion channel) and which relates to cooperativity of conformational switching (*Einav and Phillips, 2017*; *Martins and Swain, 2011*; *Marzen et al., 2013*). For plotting the three-dimensional Hill landscape ($n_H^{MWC}$) as a function of both log$L$ and log$c$, a constant $K_T$ (=143 mmHg, the average value for $K_T$ in the evolutionary dataset (See *Supplementary file 1*) and varying $K_R$ parameters were used. The Hill

biophysical landscape shape, however, is invariant to other choices made for changing the relative affinity $c$ parameter.

## MWC model simulations

Simulated MWC-based $\bar{Y}$ data were generated using *Equation 2* by changing the appropriate biophysical parameters ($K_T$, $K_R$ or $L$), as described in the text and in figure legends. For evaluating the sensitivity of oxygen unloading, simulated MWC-based $\bar{Y}$ curves were first generated around several maximum cooperativity values by systematically changing $c$ and adjusting $L$ so as to fulfill the $Lc^2=1$ maximum cooperativity criterion. Nevertheless, since scaled $\bar{Y}$ curves are presented, the analysis is invariant to parameter choice. Around each maximum cooperativity graph, the dependence of $n_H^{MWC}$ on the allosteric constant $L$ was plotted assuming the appropriate constant $c$ value in each case.

## Data extraction and fitting

To extract $\bar{Y}$ data from previously published graphs, the WebPlotDigitizer net-based program was used. Only experimental data points were sampled unless the reported graph presented the experimental trend by a continuous line rather than scattered data points, as is typical of saturation data presented in older manuscripts. In such cases, the curves were evenly sampled by taking systematic steps along the $x$-axis. $\bar{Y}$ data were fitted using either the Hill or MWC equations (*Equation 1 and 2*, respectively). The adequacy of fit was judged based on attaining a $R^2$ coefficient greater than 0.97.

## Acknowledgements

We thank Dr. Amnon Horovitz for insightful comments and to Drs. Eyal Gur, Shimon Bershtein and Jerry Eichler for critical reading of the manuscript. This work was supported by the Israel Science Foundation (grants 488/12 and 296/16 to OY).

## Additional information

### Funding

| Funder | Grant reference number | Author |
| --- | --- | --- |
| Israel Science Foundation | 488/12 | Ofer Yifrach |
| Israel Science Foundation | 296/16 | Ofer Yifrach |

The funders had no role in study design, data collection and interpretation, or the decision to submit the work for publication.

### Author contributions

Olga Rapp, Resources, Data curation, Software, Formal analysis, Validation, Investigation; Ofer Yifrach, Conceptualization, Formal analysis, Supervision, Funding acquisition, Validation, Investigation, Project administration, Writing—review and editing

### Author ORCIDs

Ofer Yifrach (iD) https://orcid.org/0000-0001-9020-9745

### Decision letter and Author response

Decision letter https://doi.org/10.7554/eLife.47640.017
Author response https://doi.org/10.7554/eLife.47640.018

## Additional files

### Supplementary files

• Supplementary file 1. MWC allosteric parameters of the hemoglobin evolutionary dataset oxygen saturation curves. [a]The MWC allosteric parameters for Hb evolutionary datasets reported in this table were obtained as described in *Rapp and Yifrach (2017)*. The $K_T$ and $K_R$ parameters

correspond to the affinity of oxygen to the **T** and **R** Hb MWC quaternary states, respectively, whereas $c$ and $L$ correspond to the relative affinity ($c = K_R/K_T$) and conformational ratio ($L = [\mathbf{T}]/[\mathbf{R}]$) of the **T** and **R** states, respectively. [b]The mammalian Hb species analyzed in the current analysis, as reported in *Milo et al. (2007)*. [c]Hill coefficient at half-saturation calculated based on *Equation 5* and using the $K_R$, $K_T$, and $L$ MWC parameters of the different mammalian Hb binding curves (see *Rapp and Yifrach, 2017*). [d]Hill coefficients at half-saturation obtained upon fitting Hb oxygen saturation data to the Hill equation (*Equation 1*), as reported in *Milo et al. (2007)*.

DOI: https://doi.org/10.7554/eLife.47640.011

• Supplementary file 2. MWC allosteric parameters of the hemoglobin physiologydataset oxygen saturation curves. [a]The table is adopted from *Rapp and Yifrach (2017)*. The MWC allosteric parameters for the Hb pH, $CO_2$ and 2,3-BPG physiological datasets were obtained using global fitting analysis, as described in *Rapp and Yifrach (2017)*. The analysis was carried out assuming a single $c$ value for each dataset curve. [b]The values for $L$ are apparent, as they were determined in the presence of varying concentrations of the same effector. [c]Hill coefficients at half-saturation calculated based on based on *Equation 5* and using the $K_R$, $K_T$, and $L$ MWC parameters of each physiological dataset curve.

DOI: https://doi.org/10.7554/eLife.47640.012

• Transparent reporting form DOI: https://doi.org/10.7554/eLife.47640.013

## Data availability

All data generated or analysed in the current study are included in the manuscript and supporting files. Supporting Files 1 and 2 delineate the principal data analyzed in the current study and were adopted from Rapp and Yifrach (2017).

The following datasets were generated:

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

## Appendix 1

DOI: https://doi.org/10.7554/eLife.47640.014

### Hb oxygen ligation is reasonably described by the MWC allosteric model

Does indeed the MWC model adequately describe Hb function? It is clear and long accepted that the MWC model sufficiently describes the cooperative binding of oxygen to Hb (*Eaton et al., 1999*). However, rationalizing the effects of modulatory ligands on Hb function using the MWC model, remained long controversial (*Imai, 1973*). This was a result of the fact that fitting saturation curves of Hb in the presence of its various inhibitors to the MWC equation, assuming $L$-only effects, consistently failed. Rather, successful fits were obtained only when both $L$ and the $c$ parameters were allowed to change (*Imai, 1973*; *Yonetani et al., 2002*). In recent years, however, it became clear that data fitting analysis using the traditional MWC model equation may not provide reliable estimates for $L$ and $c$ (*Milo et al., 2007*; *Rapp and Yifrach, 2017*) thus calling for a re-examination of previous data using alternative fitting strategies.

Recently, we reported on two simple fitting strategies that allowed obtaining reliable estimates for the $L$ and $c$ parameters of Hb saturation curves in cases of both evolutionary and physiological variations (*Rapp and Yifrach, 2017*). In particular, we showed that Hb effector dataset saturation curves may be globally fitted successfully using the MWC equation assuming varying $L$ values for the different curves and a constant $c$ value for all curves. Furthermore, we showed that the derived $L$ values of all curve scaled with effector concentration, exactly as predicted by the MWC theory. These results, thus suggested that the simple MWC model provides a reasonable description that can also account for heterotropic interactions in Hb. The tertiary two-state model (TTS) (*Viappiani et al., 2014*) is a natural extension of the MWC model, and no doubt, performs better. Still, even with this more accurate description, the traditional MWC model provides a reasonable approximation to explain steady-state physiological effects on Hb saturation. Such approximation can provide initial estimates of the 'quaternary contribution' ($L'$) of allosteric effectors within the framework of the TTS model, not previously possible, due to data fitting difficulties.

To summarize, in the analysis here we treat Hb as a pure MWC protein and make use of the values for the $L$ and $c$ MWC parameters of all evolutionary and physiological dataset saturation curves obtained using reliable data fitting strategies (*Supplementary files 1* and *2*, respectively) (*Rapp and Yifrach, 2017*), thus allowing us to address the driving forces that shaped Hb affinity and cooperativity in oxygen binding and its implications with respect to the molecular basis for body size related adaptation of Hb.

## Appendix 2

DOI: https://doi.org/10.7554/eLife.47640.014

### A good approximation for the dependence of the Hill coefficient at half saturation on the MWC model parameters

*Equation A2.1* below corresponds to *Equation 5* in the main text. As can be seen in $\bar{Y}$ MWC *Equation 2* of the main text, under half- saturation condition (where $\bar{Y} = 1/2$), $[S]_{0.5}$ is raised to a power of up to four and its explicit expression (obtained using the Mathematica software package) is long and complex. An approximation for $[S]_{0.5}$ may be obtained using the $[S]_{0.5}$ of the $\bar{R}$ conformational state function (the fraction of **R** state as a function of substrate concentration), as described in *Equation A2.2* below (*Gruber et al., 2019*). Plugging this expression into *Equation A2.1* results in an explicit expression for $n_H^{MWC}$ that depends on the MWC model parameters, as described by the long *Equation A2.3*, below. The biophysical Hill landscape obtained using this equation is very similar to that calculated directly based on *Equation A2.1* over most of the range of $L$ and $c$ values (less than 2% difference in $n_H$ values) and is identical under the parameter regime of maximum cooperativity.

$$n_H^{MWC} = \frac{\partial\left(\frac{\bar{Y}_{MWC}}{1-\bar{Y}_{MWC}}\right)}{\partial(\log[S])} = \left(4[S]\left(\frac{\partial(\bar{Y}_{MWC})}{\partial([S])}\right)\right)\Big|_{[S]=[S]_{50}} =$$

$$\left(4[S]\left(\frac{\frac{3[S]\left(1+\frac{[S]}{K_R}\right)^2}{K_R^2}+\frac{\left(1+\frac{[S]}{K_R}\right)^3}{K_R}+\frac{3L[S]\left(1+\frac{[S]}{K_T}\right)^2}{K_T^2}+\frac{L\left(1+\frac{[S]}{K_T}\right)^3}{K_T}}{\left(1+\frac{[S]}{K_R}\right)^4+L\left(1+\frac{[S]}{K_T}\right)^4} - \frac{\left(\frac{4\left(1+\frac{[S]}{K_R}\right)^3}{K_R}+\frac{4L\left(1+\frac{[S]}{K_T}\right)^3}{K_T}\right)\left(\frac{[S]\left(1+\frac{[S]}{K_R}\right)^3}{K_R}+\frac{L[S]\left(1+\frac{[S]}{K_T}\right)^3}{K_T}\right)}{\left(\left(1+\frac{[S]}{K_R}\right)^4+L\left(1+\frac{[S]}{K_T}\right)^4\right)^2}\right)\right) \quad \text{(A2.1)}$$

$$\Big|_{[S]=[S]_{50}}$$

$$[S]_{0.5} = \frac{\left(\sqrt[4]{L}-1\right)}{\frac{1}{K_R}+\frac{\sqrt[4]{L}}{K_T}} \quad \text{(A2.2)}$$

$$n_H^{MWC} = \frac{4(K_T+\sqrt[4]{L}K_R)^4(\sqrt[4]{L}^3 K_R^4(K_R+K_T)^3+K_T^4(2\sqrt[4]{L}K_R+K_T-K_R)^3)}{K_R^9 K_T^9(\sqrt[4]{L}-1(K_T+\sqrt[4]{L}K_R)\left(\frac{L(K_R+K_T)^4}{K_T^4}+\frac{(2\sqrt[4]{L}K_R+K_T-K_R)^4}{K_R^4}\right)^2} -$$

$$\frac{4\sqrt[4]{L}K_R K_T(K_T+\sqrt[4]{L}K_R)^2(\sqrt[4]{L}^3 K_R^4(K_R+K_T)^3+3K_R^4}{}$$

$$(K_R+K_T)^2(K_T+\sqrt[4]{L}K_R)-3K_T^4 L(K_T+K_R)$$

$$(2K_R\sqrt[4]{L}+K_T-K_R)^2+\sqrt[4]{L}K_T^4 L(2\sqrt[4]{L}K_R+K_T-K_R)^3)$$

$$\frac{(LK_R^4(K_R+K_T)^4+K_T^4(2\sqrt[4]{L}K_R+K_T-K_R)^4)}{K_R^9 K_T^9(\sqrt[4]{L}-1)(K_T+\sqrt[4]{L}K_R)\left(\frac{L(K_R+K_T)^4}{K_T^4}+\frac{(2\sqrt[4]{L}K_R+K_T-K_R)^4}{K_R^4}\right)^2} \quad \text{(A2.3)}$$

