## [Decision Letter]

Thank you for submitting your article "On the driving forces that shape the characteristic Hill cooperativity value of an allosteric protein" for consideration by *eLife*. Your article has been reviewed by a Senior Editor, John Kuriyan as the Reviewing Editor, and three reviewers. The following individuals involved in review of your submission have agreed to reveal their identity: Rama Ranganathan (Reviewer #2).

The reviewers have discussed the reviews with one another and the Reviewing Editor has drafted this decision to help you prepare a revised submission.

Summary:

This paper concerns the relationship between the Hill coefficient for an allosteric system (as defined by the MWC model), and the physical parameters of the model (the equilibrium constant between states and the ligand affinities). As noted by Reviewer 1, this interesting paper is a follow up to an earlier excellent article recently published in PLOS One by the same authors in which they showed that the model of Monod-Wyman-Changeux can self-consistently describe the extant data on hemoglobin, including in the presence of a variety of effectors. The current article goes much farther now by providing an interpretative framework in terms of the "Hill landscape". The argument made is that there is a surface of n_H,_the effective Hill coefficient, as a function of the MWC parameters L and c and that hemoglobins have been tuned in evolution so as to live on a ridge on this surface.

The reviewers have raised a number of issues that should be addressed in a revised version of the manuscript. Note that no additional calculations or new work is called for. But it should be clear from the reviewer comments that a careful revision of the manuscript, with inclusion of explanatory points as well as the complete set of equations in the main text, will help make this paper more effective.

Note that one of the reviewers (Reviewer 3) questions some of the fundamental assumptions underlying this work. Please respond to these concerns as well as possible.

Reviewer 1:

Essential revisions:

1) The lack of equations in the main text. For a paper that is performing a theoretical analysis couched in the language of mathematics, the relevant equations should not be relegated to the Supplementary information. Further, there are far more equations to be included (in my opinion) than the several presented there. For example, there is an equation in Figure 3 of their PLOS One paper that gives the effective L parameter that is very important and I think most people are not aware of that is the fundamental basis of interpreting how the "bare" MWC model can be used in the context of effector molecules and why the dissociation constants K_A and K_I are immune to the effector.

2) Overarching concept – I find it is a missed opportunity that the authors do so little to talk about the generic truths they are making as opposed to the very specific comments relative to hemoglobin. Further, I challenge comments such as "Hemoglobin is the only allosteric protein with a known mechanism of action". I object to this comment on several grounds – first, these authors over and over again use the word "mechanism" without once attempting to define it (at least that I can find after reading the paper carefully twice) and second, I think this is false. People have done beautiful, deep work on ion channels, on transcription factors such as LacI, on chemotaxis receptors, etc.

3) On a related point: I find the citations of the literature quite inconsistent with my own take on the literature on the field. For example, how can the amazing paper of Martins and Swain not be cited? They carried out deep analysis on the nature of allosteric molecules. Similarly, the work of – Keymer et al., 2006, Swem et al., 2009, the first on chemotaxis receptors, the latter on quorum sensing receptors. This work would benefit from embracing some of research that these people and others (Mirny, 2010; Tu et al., 2019, Marzen et al., 2015) have been doing.

4) Abstract and beyond: I found the generally evolutionary tone of this paper to be not wholly scientific. In the abstract we have "found it evolved", "showed that" and "evolutionary tuning", all indicating that the authors are convinced that hemoglobin evolved in order to achieve some optimality. There is never any discussion of the fact that correlation does not imply causation – nor any discussion of the null hypothesis of neutrality, nor any discussion of the possible fitness landscape that would produce such optimality.

5) Introduction: – "the characteristic n_H_ values of almost all allosteric proteins rarely approaches the maximal value n". In some ways I completely disagree with this perspective. My take is that this is true for only one reason and that is because the model of the Hill function is the wrong model for those molecules. If you write the MWC model, the equation (which should be front and center in the paper – their equation 2 sort of) the "4" in the partition function in the denominator is perfectly well aligned with the fact that there are 4 binding sites for hemoglobin. If you do this same thing for cyclic nucleotide gated ion channels and fit a Hill function, you don't get the correct number of binding sites. If you use MWC, you do.

6) Though the authors are constantly talking about c and L,, shouldn't we think of the MWC model as a three-parameter model? I just don't follow their thinking on this point.

7) "the molecular phenotype of a protein" – to me, the molecular phenotype is much broader than this. IT includes the leakiness 1/(1+L), the EC50, the dynamic range as well as the Hill parameter.

8) Subsection “The biophysical Hill landscape of a tetrameric MWC allosteric protein”: "summarizes all known aspects" – As mentioned above, the known aspects of the model include Leakiness, EC50, saturation, dynamic range, effective Hill parameter, but there are other deep things such as the way Wingreen and coworkers have repeatedly shown the data collapse achieved in these models. See the Keymer et al., papers where the data collapse in chemotaxis is demonstrated

9) I am not sure what they are talking about, but I think that there are mutants of MWC proteins that do something other than change L. See the work of Daber and Lewis for an example.

10) Subsection /2The characteristic n_H_ value of hemoglobin is a maximum cooperativity point”: "very close to a value of…" given that in physics people argue about exponents of 1/3 vs. 1/2 in the context of different scaling relations, it is hard to see how 2.7 and 2 are "very close".

11) "results imply that evolutionary tuning of.…" It would be very interesting to understand more about the multi-dimensional fitness landscape that a living organism faces and how a particular piece of that ends up as a low dimensional fitness landscape that only cares about n_H_. My point is that the organism as a whole faces the battle for existence. This depends upon all of its physiology, not just oxygen binding. Do all the separate pieces of physiology each optimize separately? Are some unique and privileged? The authors could attempt to address these kinds of issues rather than just giving the usual arguments about evolutionary tuning and optimization.

12) Subsection “Datasets”, "considered part" some discussion of the logic of which parts are included would be great.

13) Supplementary file 1. One thing I wondered about both here and in the PLOS paper (which I have also read many times) is about the "sloppy" nature of the parameters. I am not sure if these authors have encountered the work of Jim Sethna (Cornell, Physics), but they have done very interesting things about parameter nonuniqueness. I am not sure how that plays out here but in our own experience using MWC in many contexts, the nonuniqueness of parameters comes up again and again.

*Reviewer 2:*

This paper focuses on developing a general model that explains the mechanistic origin of the cooperativity displayed by allosteric proteins, and presents this model in the context of hemoglobin, a classic model system with the kind of diverse datasets required for this analysis. The authors claim (1) that they have rich datasets regarding oxygen saturation curves for 14 different mammalian hemoglobins in a similar physiological condition (the evolutionary dataset) and for human hemoglobin in 26 different physiological conditions (the physiological dataset), (2) that from these datasets they can derive half maximal saturation and the Hill coefficient (*n*_H_), and can derive the key parameters of the MWC model (L, the equilibrium constant of the T to R conformational change, and c, the ratio of binding affinity to the T and R states), (3) that using the so-called "Hill transformation" they can create an analytic relationship that shows by *n*_H_ depends on the mechanistic parameters L and c (a "biophysical landscape"), (4) that slices through this landscape qualitatively account for many prior reports on the literature, (5) that despite considerable variation in the actual value, the characterstic *n*_H_ of all the mammalian hemoglobins occur near to the maximum value possible given their mechanistic parameters, (6) that for physiological variations of human hemoglobin where c is fixed, all the data cluster around the maximum *n*_H_, indicating that the degree of cooperativity is rather robust to physiological changes, a property previously described as "buffering of cooperativity", (7) that adopting the maximum *n*_H_ possible facilitates the delivery of oxygen to tissues, with the buffering property ensuring that the delivery is robust to changes in physiological conditions, (8) that animals with lower body weight and higher oxygen consumption also linearly show higher maximum cooperativities, a finding that suggests evolutionarily selection of *n*_H_ for optimal organismal fitness and physiological robustness. From all this the authors conclude that they have a new global understanding of the mechanistic features that control the physiological and evolutionary dynamics of hemoglobin. By extension, there is the idea that this same framework might be useful for the study of multisubunit allosteric proteins in general (e.g. GroEL or K^+^ ion channels).

In general, this is a great paper. The logic is clear, the specific analyses and data presented do support the claims, and the general conclusions are of great interest. From this reviewer's point of view, the real value here lies in two findings: (1) that there is a low-dimensional feature (the ridge of maximum cooperativity) in the biophysical landscape in which the physiological and evolutionary diversity of hemoglobin lives, and (2) that using this landscape it is possible to argue for a logical evolutionary strategy that underlies the observed variation. These are non-trivial achievements; there are probably 10's of thousands of papers relating the various details of hemoglobin mechanism n various conditions but few that present a unifying picture. This work will be of great value to both mechanistic biochemists and evolutionary biologists.

*Reviewer 3:*

The manuscript by Rapp and Yifrach attempts to understand the mechanistic forces that guide evolutionary and physiological adaptations which determine the value of Hill coefficients in cooperative systems. Assuming a MWC model of four identical oxygen-binding sites, the authors plot *n*_H_ as a function of L and c, assuming a homogeneous tetramer. They then map published hemoglobin measurements of *n*_H_, L and c. They argue that the published *n*_H_ values lie along the maximal possible theoretical values. They also observe a linear correlation between log(L) and log[c] and suggest that this reflects some co-evolution. They further argue that their slope of the log(L)/log[c] correlation, 2.7, is very close to the maximum value allowed (2) and therefore hemoglobins have evolved to display maximal cooperativity.

I am personally not convinced regarding the validity of the claims of maximal cooperativity in hemoglobin evolution. First of all, the authors do not define maximal cooperativity. Is the value of the slope of the Hill plot at 50% saturation necessarily synonymous with (degree of) cooperativity? Second, L, c and *n*_H_ are mechanistically correlated by theory, and the correlation observed need not be taken as evidence of evolutionary cause and effect.

Essential revisions:

1) Hemoglobin is not a good model for a classical MWC-type cooperative tetramer. Many models will reproduce cooperative binding isotherms faithfully, even if they are physically implausible. A case in point is the original Hill plot itself. MWC-type cooperative binding assumes N indistinguishable binding sites, and hemoglobin has physically distinct α and β subunits whose oxygen binding sites are known to have different binding constants. The implication of two distinguishable classes of binding sites is that the maximal *n*_H_ is- a priori- lower than 4. The fact that the authors do not take this into account is problematic, as they are focusing on understanding the reasons for *n*_H_ not achieving its maximal value of 4.

2) Subsection “The characteristic *n*H value of hemoglobin is a maximum cooperativity point” and onward. The authors regenerate *n*_H_ from calculated L and c values of different hemoglobins. However, the original datasets show significantly diverging empirical *n*_H_ values. How good is the correspondence between the calculated and empirical *n*_H_ values? Does this indicate a problem with the applicability of the model?

3) Subsection “The characteristic *n*_H_ value of hemoglobin is a maximum cooperativity point” and Figure 2B. L and c are not linearly correlated. logL and log[c] are linearly correlated. I am also not sure why this indicates co-evolution, because the two parameters are linked by theory.

4) Subsection “The characteristic *n*_H_ value of hemoglobin is a maximum cooperativity point” and Figure 2B: The original formula for the relationship between L and c (from Rubin and Changeux) is L=c⋀-(n/2), where for the case of hemoglobin n=4. The authors find n/2=2.7 (Figure2B). Does this imply n=5.4? What does this imply in terms of the physical reality? The authors state that this is very close to the maximal value of 2. I am not sure that I agree, since the scale is logarithmic, base 10. Finally, does the regression line in Figure 2B cross the origin? This is not evident from the graph.

5) Figure 3A and related text. It is unclear where the authors derive their justification for invariant c values. I can see perhaps how this could, perhaps, be valid for WT hemoglobins but I cannot see any justification for constant c in the mutants.

6) The authors fail to relate the physical significance of the slope of the Hill plot to the 'degree of cooperativity' and fail to cite literature on this subject.

7) The Title is misleading and needs to explicitly state that the analysis is limited to MWC type cooperativity

8) In some cases, the citation of Levitzki, (1978) is misapplied. For example, Wyman, (1963,1964) should be cited for the modern equation defining the slope of the Hill plot (which the authors incorrectly term "Hill transformation").

---

## [Author Response]

Reviewer 1:Essential revisions:1) The lack of equations in the main text. For a paper that is performing a theoretical analysis couched in the language of mathematics, the relevant equations should not be relegated to the Supplementary information. Further, there are far more equations to be included (in my opinion) than the several presented there. For example, there is an equation in Figure 3 of their PLOS One paper that gives the effective L parameter that is very important and I think most people are not aware of that is the fundamental basis of interpreting how the "bare" MWC model can be used in the context of effector molecules and why the dissociation constants K_A and K_I are immune to the effector.

The reviewer is right. All equations are now indicated in the main text body (Results section), including the specific equation mentioned by the reviewer (now Equation 3).

2) Overarching concept – I find it is a missed opportunity that the authors do so little to talk about the generic truths they are making as opposed to the very specific comments relative to hemoglobin. Further, I challenge comments such as "Hemoglobin is the only allosteric protein with a known mechanism of action". I object to this comment on several grounds – first, these authors over and over again use the word "mechanism" without once attempting to define it (at least that I can find after reading the paper carefully twice) and second, I think this is false. People have done beautiful, deep work on ion channels, on transcription factors such as LacI, on chemotaxis receptors, etc.

We now discuss the implications and generality of our findings for understanding the molecular evolution of protein properties in subsection “How general are our findings regarding maximum cooperativity in ligand

binding?” and subsection “The relation between physiological and evolutionary adaptations in shaping a protein molecular property”. Hemoglobin as a rare protein with a known mechanism: Of course, the reviewer is right. Hb is definitely not the only protein with a known mechanism of action. In our original submission, we wrote that “Hb is the only allosteric protein with a known mechanism and for which extensive physiological and evolutionary datasets are available”. The references are given after the first half of the sentence to separate the two parts of the sentence, which is the source for the misunderstanding. Mechanism: In using the term ‘mechanism’ in the original submission, we were referring to the biophysical allosteric model that describes the ligation pathway of the protein. This was not properly explained in the original submission and is explicitly defined in the Introduction of the new submission.

3) On a related point: I find the citations of the literature quite inconsistent with my own take on the literature on the field. For example, how can the amazing paper of Martins and Swain not be cited? They carried out deep analysis on the nature of allosteric molecules. Similarly, the work of – Keymer et al., 2006, Swem et al., 2009, the first on chemotaxis receptors, the latter on quorum sensing receptors. This work would benefit from embracing some of research that these people and others (Mirny, 2010; Tu et al., 2019, Marzen et al., 2015) have been doing.

We thank the reviewer for referring us to the indicated papers. We now cite the Keymer et al., (2006), Swem et al., (2008) and Mello and Tu, (2005) papers on the introduction section as examples where the mechanism underlying protein molecular adaptation is understood. The Marzen et al., (2013) and Martins and Swain, (2011) excellent papers are also cited in the appropriate place at the main text. It should be emphasized that in these two latter papers, the expressions derived for nHMWC relate not to cooperativity in ligand binding (as derived here based onY-) but rather to cooperativity in conformational switching as it is based on the R state function (R-). This is now indicated in the subsection “Datasets”.

4) Abstract and beyond: I found the generally evolutionary tone of this paper to be not wholly scientific. In the abstract we have "found it evolved", "showed that" and "evolutionary tuning", all indicating that the authors are convinced that hemoglobin evolved in order to achieve some optimality. There is never any discussion of the fact that correlation does not imply causation – nor any discussion of the null hypothesis of neutrality, nor any discussion of the possible fitness landscape that would produce such optimality.

We concur with the reviewer. Throughout the new version, we have attempted to put forward evolutionary hypotheses, to provide evidence to substantiate them and to draw immediate conclusions and interpretations based on the data.

5) Introduction: – "the characteristic n_H_ values of almost all allosteric proteins rarely approaches the maximal value n". In some ways I completely disagree with this perspective. My take is that this is true for only one reason and that is because the model of the Hill function is the wrong model for those molecules. If you write the MWC model, the equation (which should be front and center in the paper – their equation 2 sort of) the "4" in the partition function in the denominator is perfectly well aligned with the fact that there are 4 binding sites for hemoglobin. If you do this same thing for cyclic nucleotide gated ion channels and fit a Hill function, you don't get the correct number of binding sites. If you use MWC, you do.In our original submission, we did not assign the *n*_H_ index to mean a number of subunits but rather a measure for the degree of coupling/cooperativity between subunits. Apparently, we were not clear about this. In any case, in light of the changes made in the revised version, such clarification is no longer relevant.6) Though the authors are constantly talking about c and L,, shouldn't we think of the MWC model as a three-parameter model? I just don't follow their thinking on this point.We only considered the *c* and *L* parameters for several reasons: First, according to the MWC model, cooperativity in ligand binding (*n*_H_) is determined by the *c* and *L* dimensionless parameters. *n*_H_ value is sensitive to the ratio of affinities (*c*) and not for the absolute *K*_R_ or *K*_T_ values giving rise to this ratio. Second, the dimensionless form of the MWC equation, where substrate concentration is expressed in *K*_R_ units, allows for better comparison of differences in Y-shape due to parameter choice. Three, our choice pays tribute to the original 1965 and 1966 MWC papers and other classical publications, all of which were based on use of the dimensionless form of the MWC equation. Last, it is more simple to graph and comprehend a 3- rather that a 4-dimensional surface. All but the last of these arguments are discussed at the appropriate place in the text (particularly in the subsection “The biophysical Hill landscape of a tetrameric MWC allosteric protein”) and subsection “The maximum Hill point of a MWC-type protein allows for gross- and fine-tuning of substrate unbinding sensitivity”.7) "the molecular phenotype of a protein" – to me, the molecular phenotype is much broader than this. IT includes the leakiness 1/(1+L), the EC50, the dynamic range as well as the Hill parameter.We concur with the reviewer regarding the broad definition of ‘phenotype’. In our manuscript we emphasized the need to consider *n*_H_, along with *p*_50_, in addressing molecular adaptation of Hb in mammals. These are two principal attributes of Hb function that control tissue oxygen delivery. It is indeed possible that other protein traits, not addressed in the current manuscript, might as well contribute to tissue oxygen unloading.8) Subsection “The biophysical Hill landscape of a tetrameric MWC allosteric protein”: "summarizes all known aspects" – As mentioned above, the known aspects of the model include Leakiness, EC50, saturation, dynamic range, effective Hill parameter, but there are other deep things such as the way Wingreen and coworkers have repeatedly shown the data collapse achieved in these models. See the Keymer et al., papers where the data collapse in chemotaxis is demonstrated.We are sorry for our apparent haughtiness. We meant that the landscape summarizes all cooperativity-related aspects of the MWC model. This is now corrected in the new version.9) I am not sure what they are talking about, but I think that there are mutants of MWC proteins that do something other than change L. See the work of Daber and Lewis for an example.

The lines of the original submission referred to by the reviewerdid not address Hb mutants but rather the effect of allosteric modulators on the *L* allosteric constant, in the framework of the MWC model. In any case, with respect to mutation affecting both *L* and *c*, see our comment to point # 5 of reviewer 3.

10) Subsection /2The characteristic n_H_ value of hemoglobin is a maximum cooperativity point”: "very close to a value of…" given that in physics people argue about exponents of 1/3 vs. 1/2 in the context of different scaling relations, it is hard to see how 2.7 and 2 are "very close".The reviewer is right. We now correct the term ‘very close’ to ‘close’, and provide the reasons for the apparent deviation of the observed value from that expected and allow the reader to judge the validity of our argument. Furthermore, as advised by reviewer 3, we now also point out that the log*c*-log*L* correlation crosses the (0,0) axis origin, as expected for the maximum cooperativity parameter criterion.11) "results imply that evolutionary tuning of.…" It would be very interesting to understand more about the multi-dimensional fitness landscape that a living organism faces and how a particular piece of that ends up as a low dimensional fitness landscape that only cares about n_H_. My point is that the organism as a whole faces the battle for existence. This depends upon all of its physiology, not just oxygen binding. Do all the separate pieces of physiology each optimize separately? Are some unique and privileged? The authors could attempt to address these kinds of issues rather than just giving the usual arguments about evolutionary tuning and optimization.

Indeed, in the original version, we over-stressed the cooperativity mindset and collapsed the entire fitness landscape on the *n*_H_ parameter. This, of course, is simple-minded. We now address and elaborate on the multidimensional aspects of the fitness landscape in subsection “The biophysical Hill surface as a low dimensional biophysical fitness landscape”.

12) Subsection “Datasets”, "considered part" some discussion of the logic of which parts are included would be great.In the current analysis we used only the subset of Milo et al., data for which we managed to obtain reliable (and physiological) estimates for *L* and *c*, as we report in our PLOS One paper. As we elaborated in this paper, our three equation system strategy applied for Milo et al. evolutionary dataset revealed non-physiological *L* and *c* values for 10 out of the 24 mammals (*L* values close to 1 and *c* values at the order of 0.2). This sub dataset was not further addressed in the PLOS one paper nor in here.13) Supplementary file 1. One thing I wondered about both here and in the PLOS paper (which I have also read many times) is about the "sloppy" nature of the parameters. I am not sure if these authors have encountered the work of Jim Sethna (Cornell, Physics), but they have done very interesting things about parameter nonuniqueness. I am not sure how that plays out here but in our own experience using MWC in many contexts, the nonuniqueness of parameters comes up again and again.The issue of MWC parameter dependence indeed comes up repeatedly. This was dealt, at least in part, in Milo et al. using the modified form of the MWC equation, on which our analysis is based. In any case, thanks for the reference to Jim Sethna’s work. We weren’t aware of it.Reviewer 3:I am personally not convinced regarding the validity of the claims of maximal cooperativity in hemoglobin evolution. First of all, the authors do not define maximal cooperativity. Is the value of the slope of the Hill plot at 50% saturation necessarily synonymous with (degree of) cooperativity? Second, L, c and n_H_are mechanistically correlated by theory, and the correlation observed need not be taken as evidence of evolutionary cause and effect.

*Lack of definition for the maximum cooperativity:* In our original submission, we explicitly defined the maximum cooperativity point as the extremum point of the bell-shaped *n*_H_-log*L* curves at any given *c* value (see Materials and methods section). This extremum point is obtained by setting ∂*n*_H_/∂log*L* as equal to zero, which gives rise to the *L=c^-^*^2^ parameter criterion underlying the maximum cooperativity point. This is now better explained in the main text body (subsection “The biophysical Hill landscape of a tetrameric MWC allosteric protein”). The slope value of the Hill plot is indeed a scaled slope of the actual ∂Y-/∂[S] relation and is considered as a cooperativity index to evaluate site-site coupling. We purposely derived the dependence of the Hill value at half-saturation on MWC model parameters because this is what we usually assess experimentally using the Hill equation (see also our remark in the Materials and methods section).

*Inherent dependence of L and c:* The *L* and *c* parameters of the MWC model are independent parameters. The logarithms of the values obtained for a series of proteins are not necessarily expected to be linearly dependent. Given that the experimentally derived values for *L* and *c* are reliable and were obtained using data fitting strategies preventing potential parameter dependence artifacts, we interpreted this correlation as reflecting co-evolution or evolutionary tradeoff between binding and conformational changes transitions of Hb giving rise to the maximum cooperativity point. We now stress these points better in subsection “e characteristic *n*_H_ value of hemoglobin is a maximum cooperativity point” of the new submission.

Essential revisions:1) Hemoglobin is not a good model for a classical MWC-type cooperative tetramer. Many models will reproduce cooperative binding isotherms faithfully, even if they are physically implausible. A case in point is the original Hill plot itself. MWC-type cooperative binding assumes N indistinguishable binding sites, and hemoglobin has physically distinct α and β subunits whose oxygen binding sites are known to have different binding constants. The implication of two distinguishable classes of binding sites is that the maximal n_H_is- a priori- lower than 4. The fact that the authors do not take this into account is problematic, as they are focusing on understanding the reasons for n_H_not achieving its maximal value of 4.

The MWC model is thought to adequately describe Hb function, to a first approximation, and is still widely used by many researchers (see, for example, JF Storz’s papers on Hb molecular adaptations published throughout recent years). Indeed, a model is not a proof of a mechanism and several other models can account for the data. However, the simple MWC can explain many observations made with Hb and provides a quantitative framework to interpret these data. Furthermore, by using reliable data fitting strategies, we recently presented data arguing that the pure classical MWC model can explain both homotropic and heterotropic interactions in Hb. Supplementary file 1 of our submission addresses exactly this point and summarizes our arguments regarding why we think the MWC model adequately describe Hb function. Deviations and extensions from the simple two-state model are sometimes needed to account for detailed transient kinetics data (see the Eaton's group Tertiary Two States model, for instance (Viappiani etal., (2014)). With respect to the second issue raised in this comment, as we wrote in our response to comment 5 of reviewer 1, we wanted to rationalize the *n*_H_ value not as a measure of the number of subunits but rather as a parameter for evaluating the extent of cooperativity among subunits.

2) Subsection “The characteristic n_H_value of hemoglobin is a maximum cooperativity point” and onward. The authors regenerate n_H_from calculated L and c values of different hemoglobins. However, the original datasets show significantly diverging empirical n_H_values! How good is the correspondence between the calculated and empirical n_H_values? Does this indicate a problem with the applicability of the model?

The Hill values calculated based on the MWC model are linearly correlated to those observed with a slope of almost one (0.97) and *R*^2^ coefficient of 0.92 (see Figure 1A in Rapp and Yifrach, 2017 and Supplementary file 1 (compare the two right-most columns)). Also, for the physiological data, the calculated *n*_H_ values fit those observed reasonably well (see Figure 2A in Rapp and Yifrach, 2017). In any case, our analysis assuming *L*-alone effects performs much better than those calculated by others who assumed that allosteric effectors affect both *L* and *c* (Yonetani et al., 2002). In this case, strong dependence is observed between parameters (see Figure 2B in Rapp and Yifrach, 2017).

3) Subsection “The characteristic n_H_value of hemoglobin is a maximum cooperativity point” and Figure 2B. L and c are not linearly correlated. logL and log[c] are linearly correlated. I am also not sure why this indicates co-evolution, because the two parameters are linked by theory.

As for the comment on parameter dependence, see our response above.

4) Subsection “The characteristic n_H_value of hemoglobin is a maximum cooperativity point” and Figure 2B: The original formula for the relationship between L and c (from Rubin and Changeux) is L=c⋀-(n/2), where for the case of hemoglobin n=4. The authors find n/2=2.7 (Figure2B). Does this imply n=5.4? What does this imply in terms of the physical reality? The authors state that this is very close to the maximal value of 2. I am not sure that I agree, since the scale is logarithmic, base 10…Finally, does the regression line in Figure 2B cross the origin? This is not evident from the graph.

See also our response to reviewer 1, point 18. Given the error in the slope measurement and inaccuracies steaming from data compilations (typical for meta analyses), we believe that the observed 2.7 value is close to that expected (i.e., 2). Of course, there is no physical meaning for *n*=5.4. In the ideal case, we should obtain *n*=4. We specify these arguments in the main text for the readers' judgement. As for the Y intercept of the correlation in Figure 2b, indeed it crosses very close to the axis (0,0) origin point, thus further supporting the validity of our findings. This is now reported in the main text with a new version of Figure 2B where the linear regression is extrapolated towards the 0,0 axis origin point. We thank the reviewer for raising this point.

5) Figure 3A and related text. It is unclear where the authors derive their justification for invariant c values. I can see perhaps how this could, perhaps, be valid for WT hemoglobins but I cannot see any justification for constant c in the mutants.

Hb mutations can affect either *L* or *c* or both MWC parameters. The assumption for a constant *c* value for the reported Hb mutants used in the current analysis was made by Baldwin in his original paper (1975) and further adopted by Fersht, (1985). We only follow their lead. This is now indicated in subsection “The characteristic *n_H_*value of hemoglobin is a maximum cooperativity point”).

6) The authors fail to relate the physical significance of the slope of the Hill plot to the 'degree of cooperativity' and fail to cite literature on this subject.

In the Introduction of the original submission, we explicitly wrote that the slope of the saturation curve, as evaluated by the Hill equation, is a cooperativity index to evaluate coupling between the multiple ligand binding sites. We further cited Hill’s 1910 paper.

7) The Title is misleading and needs to explicitly state that the analysis is limited to MWC type cooperativity

The Title has been changed in the revised version.

8) In some cases, the citation of Levitzki, (1978) is misapplied. For example, Wyman, (1963,1964) should be cited for the modern equation defining the slope of the Hill plot (which the authors incorrectly term "Hill transformation").

Indeed, the reviewer is right. We have added the appropriate 1964 paper of Wyman.